# Rethinking Layer-wise Model Merging through Chain of Merges

## Abstract

Fine-tuning pretrained models has become a standard pathway to achieve state-of-the-art performance across a wide range of domains, leading to a proliferation of task-specific model variants. As the number of such specialized models increases, merging them into a unified model without retraining has become a critical challenge. Existing merging techniques operate at the level of individual layers, thereby overlooking the inter-layer dependencies inherent in deep networks. We show that this simplification leads to distributional mismatches, particularly in methods that rely on intermediate activations, as changes in early layers are not properly propagated to downstream layers during merging. We identify these mismatches as a form of *internal covariate shift*, comparable to the phenomenon encountered in the initial phases of neural networks training. To address this, we propose *Chain of Merges* (CoM), a layer-wise merging procedure that sequentially merges weights across layers while sequentially updating activation statistics. By explicitly accounting for inter-layer interactions, CoM mitigates covariate shift and produces a coherent merged model through a series of conditionally optimal updates. Experiments on standard benchmarks demonstrate that CoM surpasses state-of-the-art performance. Codebase available in the suppl. material.

## 1 Introduction

The availability of large-scale pretrained models has reshaped machine learning (Radford et al., 2021; Touvron et al., 2023), with fine-tuning emerging as the most accessible path to obtaining state-of-the-art performance across diverse domains (Raffel et al., 2020; Wang et al., 2019). As these foundation models are increasingly adapted to specialized tasks and datasets, a natural question arises: *how can we combine task-specific checkpoints without retraining?* This challenge, broadly referred to as model merging (Ilharco et al., 2023; Matena & Raffel, 2022; Wortsman et al., 2022a), has recently proven effective for achieving modularity, knowledge reuse, and efficient deployment.

Since specialized modules are typically trained independently, there is no guarantee that their weights can be seamlessly combined (Yadav et al., 2023; Stoica et al., 2023). In practice, naive strategies such as weight averaging (McMahan et al., 2017; Wortsman et al., 2022a) often lead to strong performance degradation when combining heterogeneous models (Tang et al., 2024; Daheim et al., 2024; Tam et al., 2024). To tackle this challenge, the literature has proposed a wide range of heuristics, spanning techniques that mitigate interference (Yadav et al., 2023; Yu et al., 2024), align parameters via permutation-based matching (Ainsworth et al., 2023; Singh & Jaggi, 2020), preserve important weights (Matena & Raffel, 2022; Lee et al., 2025), and perform interpolation within orthonormal or task-adaptive subspaces (Marczak et al., 2025; Gargiulo et al., 2025; Tam et al., 2024). While these approaches achieve decent performance, they typically rely on problem-specific assumptions and extended hyperparameter search, lacking a unifying theoretical foundation.

A different line of work focuses on aligning model activations at the layer level (Stoica et al., 2023; Jin et al., 2023; Tatro et al., 2020; Jordan et al., 2022), typically by permuting or modifying parameters to facilitate compatibility. While these approaches lay a foundation for more principled model composition, they overlook a key challenge: layers in deep networks are not independent, but conditioned on the outputs of preceding computation. Merging them independently can introduce inconsistencies across the network. In fact, modifying early-layer parameters through merging can shift the distribution of their output activations, resulting in unexpected inputs for downstream lay-

ers. This triggers a butterfly effect, where even small discrepancies accumulate as they propagate through the network, leading to escalating mismatches and consequent performance degradation.

We identify this issue as a form of *internal covariate shift* (ICS) (Ioffe & Szegedy, 2015), a well-known problem in training dynamics where rapidly shifting early-layer activations produce unstable output distributions that hinder downstream learning (Arpit et al., 2016). Analogously, we refer to its manifestation in model merging as *merging covariate shift* (MCS), which occurs when an early layer is altered through merging, causing abrupt shifts in the inputs to subsequent layers. While merging covariate shift may be tolerable for methods operating entirely in parameter space – since they adjust weights directly without depending on intermediate activations – it becomes a critical issue for methods that rely on activation statistics. These approaches rely on layer inputs, which inevitably change when preceding layers are merged. Yet, they merge all layers simultaneously using activations computed before merging, failing to account for the resulting distributional shifts and consequently undermining performance.

To address this challenge, we propose **Chain of Merges (CoM)**: a recursive merging approach that begins at the input layer and iteratively updates parameters until reaching the last one. Specifically, we propose updating activation statistics after each merging step, replacing the original task-specific activations with those produced by the partially merged model. This process explicitly captures inter-layer dependencies and ensures global consistency, providing a framework applicable to any activation-based merging methodology. Building on this, we follow Jin et al. (2023) and cast parameter merging as a layer-wise distillation problem, where the merged weights are optimized to replicate the activation distributions of the original task-specific modules. This problem admits a closed-form solution for linear layers, which constitute a substantial portion of transformer-based architectures and are typically the only layers optimized during fine-tuning (e.g., LoRA-style adaptation keeps all other weights fixed (Hu et al., 2022)).

Our main contributions can be summarized as follows:

- We identify and analyze the presence of internal covariate shift, empirically showing that activation mismatches accumulate across layers.
- We introduce Chain of Merges (CoM), a novel merging technique that progressively distills parameters by updating activation statistics, ensuring consistency across the network.
- We evaluate CoM on standard model merging benchmarks across language and vision settings, showing it outperforms existing methods by a large margin on LoRA fine-tuning and provides state of the art performance on traditional fine-tuning.

## 2 BACKGROUND AND MOTIVATION

Model merging aims to combine a collection of $N$ models, all sharing an identical architecture, independently trained on distinct input datasets. Each model comprises $L$ linear layers, which are the target of the merging procedure. For a given layer $l \in \{1, 2, \ldots, L\}$, the set of corresponding weight matrices is denoted as $\{\boldsymbol{W}_i^l\}_{i=1}^N$, all having the same dimensions.

RegMean (Jin et al., 2023) proposes to find a single linear transformation, $\boldsymbol{W}_M^l$, that best approximates the behavior of the original layers when applied to their respective inputs $\{\boldsymbol{X}_i^l\}_{i=1}^N$. This is accomplished by minimizing the following objective function:

$$\text{minimize } \Omega^l = \sum_{i=1}^N \left\| \boldsymbol{W}_M^l \boldsymbol{X}_i^l - \boldsymbol{W}_i^l \boldsymbol{X}_i^l \right\|_2^2. \tag{1}$$

By differentiating $\Omega^l$ with respect to $\boldsymbol{W}_M^l$ and setting the result to zero, we can obtain a closed-form solution for the optimal merged layer:

$$\boldsymbol{W}_M^l = \left( \sum_{i=1}^N \boldsymbol{W}_i^l \boldsymbol{X}_i^l \boldsymbol{X}_i^{l\top} \right) \left( \sum_{i=1}^N \boldsymbol{X}_i^l \boldsymbol{X}_i^{l\top} \right)^{-1}. \tag{2}$$

Here, each $\mathbf{X}_i^l \in \mathbb{R}^{d \times \text{samples}}$ is the input data to the $l^{\text{th}}$ layer of the $i^{\text{th}}$ model, and the corresponding Gram matrix $\boldsymbol{X}_i^l \boldsymbol{X}_i^{l\top} \in \mathbb{R}^{d \times d}$ captures the pairwise correlations between individual examples.

## 3 METHODOLOGY

**Merging Covariate Shift.**    When using Equation 2, the resulting $W_M^l$ closely matches the outputs of the original task-specific layers in isolation. However, merging all layers simultaneously based on the initial inputs overlooks the dependencies between successive layers. Specifically, the inputs $X_i^l$ to the $l^{th}$ layer in Equation 2 correspond to the activations of the $(l-1)^{th}$ layer. Once the original parameters $W_i^{l-1}$ are replaced with their merged counterparts $W_M^{l-1}$, these activations shift accordingly. As a result, the statistics used to merge layer $l$ no longer align with the distribution actually produced after layer $(l-1)$ has been merged. This mismatch induces a shift in activation statistics, analogous to internal covariate shift, which we refer to as merging covariate shift (MCS).

### 3.1 CHAIN OF MERGES

**Recursive Dependence.**    To address MCS, we revise the closed-form solution presented in Equation 2. Instead of relying on the inputs $X_i^l$ – that is, the activations produced by the preceding **unmerged** layer – we employ the activations produced by the preceding **merged** layer, $\hat{X}_i^l$. These represent the actual inputs received by layer $l$ during inference, once all preceding layers have been merged. Formally, we define the pre- and post-merging inputs to layer $l$ of model $i$ as:

$$X_i^l \;=\; \sigma^{l-1}\left(W_i^{l-1}X_i^{l-1}\right), \qquad \text{and} \qquad \hat{X}_i^l \;=\; \sigma^{l-1}\left(W_M^{l-1}\hat{X}_i^{l-1}\right), \qquad (3)$$

where $\sigma^l$ denotes the (possibly composite) activation function connecting layers $l$ and $(l+1)$. Substituting $X_i^l$ for $\hat{X}_i^l$ yields the revised expression for the merged weights:

$$W_M^l = \left(\sum_{i=1}^N W_i^l \hat{X}_i^l \hat{X}_i^{l\top}\right)\left(\sum_{i=1}^N \hat{X}_i^l \hat{X}_i^{l\top}\right)^{-1}. \qquad (4)$$

This substitution induces a *recursive* dependence: the inputs to layer $l$ now depend on the outputs of layer $(l-1)$, which are themselves computed using the merged weights $W_M^{l-1}$ and the inputs to layer $(l-1)$. In turn, these depend on the merged outputs of layer $(l-2)$, and so forth. As a result, at every preceding layer all unmerged activations and parameters must be replaced by their merged counterparts, propagating the correction backward through the entire network.

**Initial step.**    The recursive chain starts at the point where either the weights or the inputs are fixed. Such a base case occurs at the first layer, where the inputs correspond to raw data and are unaffected by prior merging. Consequently, the merged weights can be directly computed using Equation 2 as:

$$W_M^1 = \left(\sum_{i=1}^N W_i^1 X_i^1 X_i^{1\top}\right)\left(\sum_{i=1}^N X_i^1 X_i^{1\top}\right)^{-1}. \qquad (5)$$

**Recursive step.**    The merged weights from the initial step are used to propagate consistent activations forward through the network. For subsequent layers $l = 2, \dots, L$, the algorithm proceeds recursively, alternating between computing the activations $\hat{X}_i^l$ and updating the merged weights $W_M^l$ according to Equation 4. By ensuring that the computed statistics at each stage reflect the accumulated effect of all preceding merges, this auto-regressive scheme *fully mitigates* merging covariate shift throughout the network. Importantly, this recursive formulation incurs no extra cost beyond the computation required for Equation 4, as discussed in Section C.

### 3.2 GRAM-BASED IMPORTANCE

Although the merged weight matrix produced by our strategy is *optimal* with respect to the regression objective, it must be regarded as an approximation rather than an exact reconstruction of the original task-specific weights. This limitation comes from the structural compression of multiple linear transformations into a single one of fixed dimensionality, which necessarily discards some representational capacity. As a result, a residual regression error is *inevitable*, introducing perturbations into the activations of the merged model across all tasks.

**Task Importance.**    The objective of model merging is to retain the performance across all tasks; however, the relative importance of each task is not uniform. Tasks that are semantically similar to the pretraining naturally benefit from the representations of the base model, as their data

distribution aligns with that seen during pretraining. In contrast, those that are semantically distant cannot rely on pretrained features and face a higher risk of performance degradation after merging. Preserving these tasks is more critical, as they stand to lose the most if their task-specific weights are not adequately incorporated into the merged model. To account for this asymmetry, we assign each task–layer pair an importance weight $\omega_i^l$, which should reflect how strongly the merged weights have to be biased toward preserving task $i$ when merging layer $l$. The key challenge is to derive a layer-wise surrogate measure for this importance without relying on downstream evaluation data.

---

**Algorithm 1** – Chain of Merges

**Require:** Model weights $\{\boldsymbol{W}_i^l\}_{i=1,2,\ldots,N}^{l=1,2,\ldots,L}$, first layer inputs $\{\boldsymbol{X}_i^1\}_{i=1}^N$, depth $L$

1: **for** $l = 1$ to $L$ **do**
2:    **for** $i = 1$ to $N$ **do**
3:       **if** $l = 1$ **then**
4:          $\hat{\boldsymbol{X}}_i^1 \leftarrow \boldsymbol{X}_i^1$
5:       **else**
6:          $\hat{\boldsymbol{X}}_i^l \leftarrow \sigma^{l-1}(\boldsymbol{W}_M^{l-1}\hat{\boldsymbol{X}}_i^{l-1})$
7:       $\tilde{\boldsymbol{X}}_i^l \leftarrow \hat{\boldsymbol{X}}_i^l \,/\, \|\hat{\boldsymbol{X}}_i^l\|_2$
8:       $\boldsymbol{G}_i^l \leftarrow \tilde{\boldsymbol{X}}_i^l \tilde{\boldsymbol{X}}_i^{l\top}$
9:       $\omega_i^l \leftarrow \sum_{p < q} |(\boldsymbol{G}_i^l)_{pq}|$
10:   $\boldsymbol{W}_M^l \leftarrow \left(\sum_{i=1}^N \omega_i^l \boldsymbol{W}_i^l \boldsymbol{G}_i^l\right)\left(\sum_{i=1}^N \omega_i^l \boldsymbol{G}_i^l\right)^{-1}$
11: **return** $\{\boldsymbol{W}_M^l\}_{l=1}^L$

---

**Cosine similarity as a proxy.** To quantify a task's semantic distance from pretraining, we use the overall cosine similarity of the layer's pretrained input features $\boldsymbol{X}_i^l$, which reflects redundancy in task-specific representations. Highly correlated features align in similar directions, deviating from the isotropic structure of pretraining, and indicating greater task importance as the task is harder to preserve. In contrast, weakly correlated features suggest a distribution that is closer to pretraining, making the task easier to preserve. This is consistent with prior work (Cogswell et al., 2015; Morcos et al., 2018), demonstrating that decorrelated features enhance generalization, and empirically validated in Section 5.

In practice, we use the merged model's features rather than the pretrained ones, as our merging procedure already computes the Gram matrix of each layer's inputs $\boldsymbol{G}_i^l$ (Equation 4). Specifically, when constructing $\boldsymbol{G}_i^l$, we L2-normalize the input features $\hat{\boldsymbol{X}}_i^l$ to effectively produce a cosine similarity matrix, and define task importance as the absolute sum of the off-diagonal terms of $\boldsymbol{G}_i^l$:

$$\omega_i^l = \sum_{p<q} |(\boldsymbol{G}_i^l)_{pq}|, \qquad \text{with } \boldsymbol{G}_i^l = \tilde{\boldsymbol{X}}_i^l \tilde{\boldsymbol{X}}_i^{l\top}, \qquad \tilde{\boldsymbol{X}}_i^l = \frac{\hat{\boldsymbol{X}}_i^l}{\|\hat{\boldsymbol{X}}_i^l\|_2}. \tag{6}$$

This metric measures the overall cosine similarity between intermediate activations. Larger $\omega_i^l$ indicates stronger correlations, meaning the task has diverged more during fine-tuning and is more critical to retain, while smaller $\omega_i^l$ reflects near-orthogonal features, suggesting the task is semantically close to the pretrained model.

**Activation Normalization.** Our approach requires the inversion of $\sum_{i=1}^N \hat{\boldsymbol{X}}_i^l \hat{\boldsymbol{X}}_i^{l\top}$, whose conditioning critically affects the numerical stability of the solution. In Transformer architectures, activation Gram matrices are frequently ill-conditioned due to the inherent low-dimensionality of the underlying token representations (Barbero et al., 2024; Arefin et al., 2024), especially as fine-tuning typically occurs within a low-dimensional subspace (Aghajanyan et al., 2021; Kumar et al., 2022). Drawing on prior work showing that layer normalization (Ba et al., 2016) mitigates representational collapse (Wu et al., 2024), we replace the gram matrix $\hat{\boldsymbol{X}}_i^l \hat{\boldsymbol{X}}_i^{l\top}$ with the cosine similarity matrix $\boldsymbol{G}_i^l$, which is computed from normalized features and therefore better conditioned. The complete procedure of the proposed methodology is summarized in Algorithm 1.

**Importance-weighted merging.** Finally, to incorporate task- and layer-specific importance, we extend our objective $\Omega$ (Equation 1) with the weighting factor $\omega_i^l$, biasing the merged layer toward more sensitive tasks. The resulting merging rule for $\boldsymbol{W}_M^l$ is given by:

$$\underset{\boldsymbol{W}_M^l}{\arg\min} \sum_{i=1}^N \omega_i^l \left\| \boldsymbol{W}_M^l \tilde{\boldsymbol{X}}_i^l - \boldsymbol{W}_i^l \tilde{\boldsymbol{X}}_i^l \right\|_2^2 = \left(\sum_{i=1}^N \omega_i^l \boldsymbol{W}_i^l \boldsymbol{G}_i^l\right)\left(\sum_{i=1}^N \omega_i^l \boldsymbol{G}_i^l\right)^{-1}. \tag{7}$$

The complete procedure of the proposed method is summarized in Algorithm 1, with the full derivation of Equation 7 provided in Section A.

# 4 EXPERIMENTAL STUDY

## 4.1 EVALUATION SETTINGS

**Vision experiments.** We evaluate our approach in the vision domain using the benchmark of Ilharco et al. (2023), which involves merging checkpoints from eight classification datasets: Stanford Cars (Krause et al., 2013), DTD (Cimpoi et al., 2014), EuroSAT (Helber et al., 2019), GTSRB (Stallkamp et al., 2011), MNIST (LeCun et al., 2002), RESISC45 (Cheng et al., 2017), SUN397 (Xiao et al., 2016), and SVHN (Netzer et al., 2011).

**Language experiments.** Following Stoica et al. (2025); Panariello et al. (2025), we assess model generalization to the language domain on six datasets: SNLI (Bowman et al., 2015), MultiNLI (Williams et al., 2017), SICK (Marelli et al., 2014), SciTail (Khot et al., 2018), RTE (Wang et al., 2019), and QNLI (Wang et al., 2019). In SNLI, MultiNLI, and SICK, the task is to classify the relationship between a *premise* and a *hypothesis* as entailment, contradiction, or neutral. SciTail, RTE, and QNLI only involve two labels, so the outputs space is restricted accordingly.

**Evaluated approaches.** We compare our method with 14 leading techniques in the model-merging domain. *Task-Arithmetic (TA)* (Ilharco et al., 2023), *TIES* (Yadav et al., 2023), and *DARE* (Yu et al., 2024) operate by directly merging task vectors in weight space. Improving on them, *Consensus TA* (Wang et al., 2024) prunes task-specific checkpoints to retain only shared parameters before merging, while *LiNeS* (Wang et al., 2025) scales parameter updates by layer depth to preserve both general features and task-specific representations. Among SVD-based techniques, *KnOTS* (Stoica et al., 2025) aligns weights to improve merging, enhancing existing methods like TIES and DARE. Instead, *TSV* (Gargiulo et al., 2025) compresses the weights and estimates task interference to guide the merging process, while *Iso-C* (Marczak et al., 2025) perform isotropization, decomposing the weights via SVD and reconstructing them with equal singular values. *Reg-Mean* (Jin et al., 2023) aligns model parameters by solving a closed-form regression problem across all linear layers, while *FisherAVG* (Matena & Raffel, 2022) combines models using Fisher Information as importance weights. *MaTS* (Tam et al., 2024) extends these two by leveraging conjugate gradient optimization to align models within their respective task-parameter subspaces. On a different line, *AdaMerging++* Yang et al. (2024) and *ProDistill* Xu et al. (2025) learn layer-wise scalar coefficients via gradient descent; the former minimizes the entropy of the final predictions and the latter reduces the $\ell_2$ norm between fine-tuned and merged layer activations. Finally, *Localize and Stitch* He et al. (2024) leverages validation data to find and keep just $1\%$ of the model's parameters, minimizing conflicts during merging. *Zero-shot* denotes CLIP's zero-shot performance, while *Individual FT* denotes the performance of each fine-tuned model when evaluated on its own.

**Evaluation Protocol.** All our experiments follow a *static model-merging* protocol: the merger outputs a single set of parameters, which is used for all tasks and all inputs at inference time. No task identifiers, routing mechanisms, or input-dependent adapters are allowed at test time. This constrasts with *dynamic* (or conditional) merging, where the parameters depend on the task or input.

**Implementation Details.** Consistent with prior works Stoica et al. (2025); Gargiulo et al. (2025); Ilharco et al. (2023), we use ViT-B/32 and ViT-L/14 (Dosovitskiy et al., 2021) CLIP encoders as the vision-task backbones for all examined methods. For natural language tasks, we utilize Llama 3-8B (Grattafiori et al., 2024). Each model is fine-tuned using LoRA (Hu et al., 2022) or traditional fine-tuning. While the former is applied solely to attention modules (*i.e.*, query, key, value, and output projection layers) with rank $r = 16$, the latter modifies all weights; for all non-linear layers, we employ simple averaging. To ensure both reproducibility and fair comparison, we employ the LoRA fine-tuned checkpoints provided by Stoica et al. (2025), and the full fine-tuning checkpoint from Ilharco et al. (2023). We the use bfloat16 data type for NLP tasks – as it was shown to generally outperform float16 (Kalamkar et al., 2019) – except during Gram-matrix inversion, where float32 is used to ensure numerical stability. Following the original benchmark, we report the average normalized accuracy of the merged model across all datasets. For constructing the Gram matrices, we draw balanced examples across tasks and classes; if the number of classes exceeds the examples, classes are randomly subsampled. To mitigate conditioning issues (see Section 3), we use the Moore–Penrose pseudoinverse together with Tikhonov regularization (Hoerl & Kennard, 1970).

Table 1: Normalized and absolute average accuracies (%) of merged models with ViT-B/32, ViT-L/14, and Llama-3 8B. Best results in bold, second-best underlined.

| Method | ViT-B/32 | | ViT-L/14 | | Llama-3 8B | |
|---|---|---|---|---|---|---|
| | Norm | Abs | Norm | Abs | Norm | Abs |
| Zero-shot | 57.49 | 48.32 | 70.11 | 64.69 | 51.09 | 47.42 |
| Individual FT | 100.0 | 84.05 | 100.0 | 92.27 | 100.0 | 92.54 |
| TA | 63.78 | 53.61 | 74.79 | 69.01 | 90.38 | 83.64 |
| TIES | 63.70 | 53.54 | 75.51 | 69.67 | 91.08 | 84.29 |
| DARE$_{TIES}$ | 63.65 | 53.50 | 75.53 | 69.69 | 89.44 | 82.77 |
| Consensus TA | 64.72 | 54.40 | 76.70 | 70.77 | 90.79 | 84.02 |
| LiNeS | 63.63 | 53.48 | 74.65 | 68.88 | 90.84 | 84.06 |
| FisherAVG | 70.04 | 54.87 | 75.32 | 69.50 | — | — |
| RegMean | 66.02 | 55.49 | 69.85 | 64.45 | 87.58 | 81.05 |
| MaTS | 70.01 | 58.84 | 75.97 | 70.10 | — | — |
| TSV | 66.66 | 56.03 | 77.99 | 71.96 | 92.55 | 85.65 |
| Iso-C | 70.66 | 59.39 | 83.70 | 77.23 | 57.08 | 52.82 |
| KnOTS$_{TIES}$ | 67.73 | 56.93 | 78.99 | 72.88 | 92.53 | 85.63 |
| **CoM (ours)** | **92.62** | **77.85** | **91.06** | **84.02** | **99.69** | **92.25** |

## 4.2 RESULTS

**Vision tasks — LoRA.** On the smaller ViT-B/32 model, simple parameter-space methods yield limited performance: TA, TIES, DARE, Consensus TA, and LiNeS reach normalized accuracies in the mid-60s, and absolute ones in the mid 50%. Instead, more advanced baselines show mixed results. RegMean, TSV, and KnOTS$_{TIES}$ provide modest improvements, while FisherAVG, MaTS, and Iso-C perform at a similar level and group in the second-best tier. CoM sharply improves performance across the board, achieving an average normalized accuracy of 92.62% and surpassing this second group by more than 20 points.

Scaling to ViT-L/14. Moving to the larger ViT-L/14 backbone lifts the performance of nearly all methods and closes the gap between simple parameter-space baselines and more advanced approaches, placing the median normalized accuracy around 75%. TSV and KnOTS gain a couple of points thanks to their SVD-based merging, and Iso-C secures the second-best results with a solid margin. In contrast, RegMean becomes an outlier, underperforming with respect to all other baselines, suggesting that activation matching is less effective than straightforward averaging on this larger architecture. Even so, CoM delivers state of the art performance with a clear gap, reaching 91.06% versus 83.70% for the second-best method, Iso-C. The sharp contrast between RegMean's performance and that of CoM suggests that Merging Covariate Shift is more pronounced in the ViT-L model than in its base variant.

**Language tasks – LoRA.** Results on the six natural language benchmarks with LLaMA 3-8B ( Table 1) show that merging is generally less destructive in this domain, as weight-space baselines such as Task Arithmetic already retain very high normalized accuracy (90.38% on average). More sophisticated parameter-based techniques provide small gains w.r.t. TA, while other approaches yield mixed results: KnOTS$_{TIES}$ and TSV perform the best, whereas RegMean drops lower than TA, following a similar trend shown with the ViT-L architecture. A clear outlier is Iso-C, which drops to 57.08%, likely because rescaling singular values interacts poorly with language model fine-tuning.

At the top, CoM delivers near-perfect merged performance, with an average normalized accuracy of 99.69%, surpassing the strongest baseline by more than 7 points. Together with the vision results, these findings indicate that CoM is a broadly effective merging strategy that best preserves the performance of specialized models. We do not report FisherAVG or MaTS, as computing the Fisher Information Matrix is prohibitively expensive for this architecture.

**Vision tasks – Full fine-tuning.** With full fine-tuning, performance improves substantially across all methods compared to LoRA. On ViT-B/32, simple parameter-space baselines such as TA and TIES already achieve competitive results, with normalized accuracies of 76.5% and 81.0%, respectively. However, here more advanced methods make a strong difference: Localize-and-Stitch and AdaMerging++ deliver strong results around the mid-to-high 80s, while ProDistill and TSV lead the baselines with normalized accuracies above 92%. CoM further advances the state of the art, achieving 94.8%/87.7% normalized and absolute accuracy.

Scaling up to the ViT-L/14 backbone strengthens the same trend. Most methods see consistent gains, with AdaMerging++, ProDistill, and TSV all surpassing 95% normalized accuracy. CoM once again achieves the top results, reaching 97.8%

Table 2: Normalized and absolute average accuracies (%) of merged models with ViT-B/32 and ViT-L/14. Best results in bold, second-best underlined.

| | ViT-B/32 | | ViT-L/14 | |
|---|---|---|---|---|
| **Method** | **Norm** | **Abs** | **Norm** | **Abs** |
| Zero-shot | 52.2 | 48.3 | 67.6 | 64.7 |
| Individual FT | 100.0 | 92.5 | 100.0 | 95.7 |
| TA | 76.5 | 70.8 | 88.7 | 84.9 |
| TIES | 81.2 | 75.1 | 90.8 | 86.9 |
| Consensus TA | 81.4 | 75.0 | 90.2 | 86.3 |
| FisherAVG | 73.8 | 68.3 | 85.9 | 82.2 |
| RegMean | 77.6 | 71.8 | 87.5 | 83.7 |
| Loc-and-Stitch | 86.3 | 79.9 | 90.4 | 86.5 |
| AdaMerging++ | 87.6 | 81.1 | 95.1 | 91.0 |
| ProDistill | 92.9 | 86.0 | 96.1 | 91.9 |
| TSV | 92.8 | 85.9 | 97.2 | 93.0 |
| **CoM (ours)** | **94.8** | **87.7** | **97.8** | **93.6** |

normalized and 93.6% absolute accuracy, setting a new performance benchmark. These results confirm that CoM remains highly effective even under full fine-tuning, producing merged models that mostly preserve the performance of their specialized counterparts. However, the smaller performance gap between CoM and competing methods suggests that closed-form activation matching is less effective when merging full-rank checkpoints.

## 5 MODEL ANALYSIS

**Impact of Individual Components.** While CoM is primarily designed to mitigate merging covariate shift during composition, it also incorporates additional components that contribute to overall performance. We quantify the contribution of each component via an ablation study, presented in Table 3, systematically removing them to assess their individual impact on performance (we report the Average column). Our results indicate that addressing merging covariate shift alone is sufficient to achieve state-of-the-art performance.

Table 3: Impact of individual components in both vision and language tasks. *MCS* denotes solving merging covariate shift, *Norm* refers to activation normalization, and *Sim* indicates Similarity-based importance weighting.

| Components | | | Architecture | | |
|---|---|---|---|---|---|
| MCS | Norm | Sim | ViT-B/32 | ViT-L/14 | Llama3-8B |
| ✗ | ✗ | ✗ | 66.02 | 69.85 | 87.58 |
| ✓ | ✗ | ✗ | 82.53 | 83.55 | 99.45 |
| ✓ | ✓ | ✗ | 83.51 | 86.80 | 99.60 |
| ✓ | ✓ | ✓ | 92.62 | 91.06 | 99.69 |

However, other components also play a significant role. In particular, weighting by the off-diagonal norm leads to substantial improvements in vision domains, while proving less critical for language tasks. We attribute this discrepancy to the inherent differences between the two settings: language tasks involve highly similar distributions, sharing the *same label space* across all datasets, whereas vision tasks correspond to classification problems defined on entirely different classes and domains. A more detailed discussion is provided in Section B. Finally, although activation normalization has a comparatively smaller impact, it consistently enhances performance across all benchmarks by ensuring numerical stability of the solution.

**CoM – Number of examples.** Since CoM leverages input data to estimate task-specific Gram matrices, we perform an ablation study to determine the amount of data required for reliable performance. Table 4 reports the influence of the number of samples used to estimate the Gram matrices on each task. A sufficiently large sample size is essential to ensure numerical stability of the matrices,

Table 4: Performance (Average column) of CoM varying the number of examples used for each task.

| # of samples | 2 | 5 | 10 | 50 | 100 | 200 | 300 | 400 | 500 |
|---|---|---|---|---|---|---|---|---|---|
| ViT-B/32 | 75.35 | 83.86 | 87.26 | 90.61 | 91.86 | 92.20 | 92.21 | 92.38 | 92.40 |
| ViT-L/14 | 81.77 | 87.21 | 88.95 | 90.57 | 90.99 | 90.97 | 90.99 | 91.00 | 91.06 |
| Llama3-8B | 96.15 | 97.03 | 98.50 | 98.62 | 98.95 | 99.04 | 99.50 | 99.48 | 99.33 |

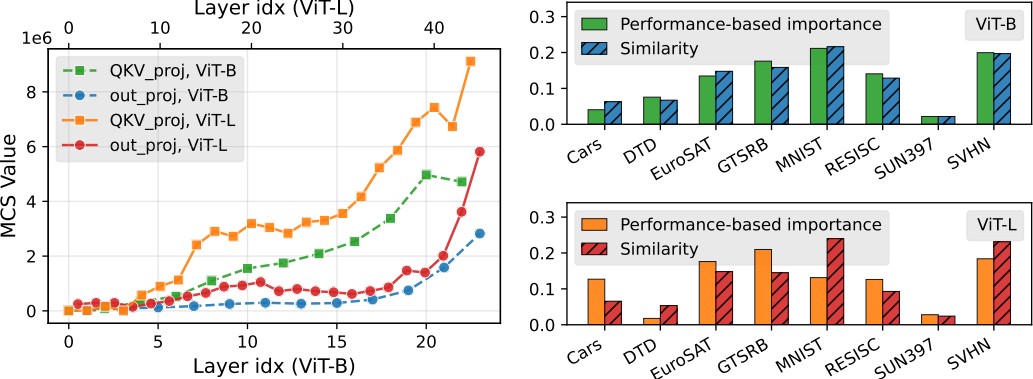

Figure 1: Merging covariate shift across layers.     Figure 2: Task importance *vs.* similarity.

which allows the inversion in Equation 2 to produce meaningful results. Empirically, we observe that stability can be maintained with as few as 2 samples per task. As shown in Table 4, CoM already surpasses the current state of the art with only 2 samples and approaches near-maximal performance with 100 samples, highlighting its data efficiency and robustness across tasks.

**Measuring MCS.** Covariate shift refers to changes in data distributions. Therefore, it is necessary to define a suitable distribution over the network activations in order to quantify this phenomenon. Following Huang & Yu (2020), we model the distribution of activation outputs as a multivariate Gaussian and evaluate MCS using the Earth Mover's (EM) Distance (Villani, 2008), also referred to as the squared 2-Wasserstein distance or the Fréchet distance (Dowson & Landau, 1982). This choice is convenient, as the EM distance admits a closed-form solution for Gaussian distributions. The total merging covariate shift for the $l^{\text{th}}$ layer can be measured as:

$$\text{MCS}^l = \sum_{i=1}^N \|\mu_{i,l} - \hat{\mu}_{i,l}\|_2^2 + \text{Tr}\left[\Sigma_{i,l} + \hat{\Sigma}_{i,l} - 2\left(\hat{\Sigma}_{i,l}^{1/2}\,\Sigma_{i,l}\,\hat{\Sigma}_{i,l}^{1/2}\right)^{1/2}\right], \tag{8}$$

where $\mu_{i,l}$ and $\hat{\mu}_{i,l}$ denote the empirical means, while $\Sigma_{i,l}$ and $\hat{\Sigma}_{i,l}$ represent the empirical covariance matrices of the inputs $\boldsymbol{X}_i^l$ and $\hat{\boldsymbol{X}}_i^l$ (Equation 3), respectively. To investigate the presence of MCS during model merging, we measure it using Equation 8 and report the results in Figure 1. The results indicate that MCS is present across all layers and tends to increase with depth, as earlier layers influence subsequent ones and the mismatch accumulates. We analyze the two projections before and after attention (the only fine-tuned layers) separately, as they show slightly different behaviors.

**Feature correlation.** To motivate similarity-weighted merging beyond its empirical effectiveness, we investigate whether the semantic distance between tasks and pretraining varies and whether it correlates with our weighting factor $\omega_i^l$. We estimate this distance using the performance gap between each task-specific model (evaluated in isolation) and the zero-shot performance of the base model, which reflects how much fine-tuning can improve the considered task. In Figure 2, we compare this accuracy gap with our similarity-based weighting factor, averaged across layers to produce a single value per task. The two measures, normalized for visual comparison, exhibit a consistent correlation across vision datasets and architectures, supporting the rationale behind our weighting scheme. Results for textual datasets are provided in Section B.

## 6 COMPARISON WITH RELATED WORK

Pioneering model merging techniques rely on linear interpolation of model parameters. FedAvg McMahan et al. (2017) introduces this approach in the context of Federated Learning, assuming a shared initialization across models. Building on this idea, Model Soups Wortsman et al. (2022a) propose a greedy strategy that incrementally incorporates models into the mixture only when they improve validation performance; WiSE-FT Wortsman et al. (2022b) enhances fine-tuning by weighting model updates to boost generalization and robustness across tasks, and Task Arithmetic Ilharco et al. (2023) enables personalized model editing, allowing finer control over individual contributions. Despite their simplicity and effectiveness, these methods can suffer performance degradation due to conflicting parameters Tang et al. (2024); Du et al. (2024); Tam et al. (2024).

To address this, a family of approaches seeks to reduce interference by applying heuristics prior to merging: TIES Yadav et al. (2023) addresses parameter redundancy by pruning and aligning parameter signs; DARE Yu et al. (2024) applies stochastic parameter dropping and rescaling; Consensus Merging Wang et al. (2024) learns task-specific masks to preserve performance with fewer parameters; LiNeS Wang et al. (2025) adjusts parameter magnitudes based on their layer depth within the network; and Git Rebasin Ainsworth et al. (2023) leverages Linear Mode Connectivity, arguing that fine-tuned models lie within the same loss basin up to a permutation of their parameters.

In parallel, another line of research argues that model ensembling should be performed within subspaces that maximize alignment between parameters. Within this framework, TSV Gargiulo et al. (2025) compresses parameters into a low-rank structure and approximates whitening by solving the Procrustes orthogonality problem. KnOTS Stoica et al. (2025) performs merging in an aligned parameter space using singular value decomposition, while ISO-C Marczak et al. (2025) enforces layer-wise isotropic matrices by rescaling singular values to produce singular vectors of equal magnitude. While these methodologies rely on heuristics applied directly to parameters, CoM aligns activations, enabling more precise model merging that optimally preserves task-specific features.

A complementary research direction focus on aligning model features to facilitate merging. Neuron Alignment Tatro et al. (2020) uses layer-wise regression to compute bipartite matches between neurons, aligning intermediate activations. ZIPIt! Stoica et al. (2023) applies intermediate feature-based permutations to align layers, while Optimal Transport Fusion Singh & Jaggi (2020) formulates alignment as an optimal transport problem, computing soft matching between activation distributions. DF-Merge Lee et al. (2025) combines parameter importance with scaling, using Bayesian optimization to tune the coefficients, while Fisher-weighted averaging Matena & Raffel (2022) directly averages parameters weighted by the Fisher Information Matrix Fisher (1922) to maximize alignment of final features. Regmean Jin et al. (2023) derives a closed-form solution to match activations layer-wise, while MaTS Tam et al. (2024) unifies the two preceding approaches under a common linear system and proposes solving it via conjugate gradient rather than in closed form. Similarly to RegMean, ProDistill minimizes the $\ell_2$ norm between fine-tuned and merged models activations, but learns layer-wise scalar coefficients via gradient descent instead of solving the objective in closed form. AdaMerging learns task- or layer-wise scalar coefficients via gradient descent, but minimizes the entropy of the final predictions; it optimizes all layers jointly, making it one of the few methods robust to MCS. In contrast, our Chain of Merges addresses these shifts explicitly by updating activation statistics sequentially, layer by layer, thereby preserving network consistency.

A complementary line of work replaces a single, task-agnostic parameter vector with input- or task-conditioned inference. EMR-Merging elects a unified base and applies lightweight task-specific masks and rescalers at test time Huang et al. (2024). Similarly, Twin-Merging modularizes knowledge into shared and exclusive components, dynamically integrating them for each input using a router module Lu et al. (2024). WeMoE, instead, merges part of the modules statically, leveraging a Mixture-of-Experts on the MLP layers only, with a routing mechanism selecting experts at inference Tang et al. (2024). These approaches operate under a dynamic merging protocol with task- or input-dependent routing, which differs from the static setting adopted in our experiments.

## 7 CONCLUSIONS AND FUTURE WORK

In this work, we identify and address Merging Covariate Shift (MCS), a form of internal covariate shift that emerges when merging layers independently in methodologies that rely on activation statis-

tics. To mitigate the adversarial effect of this phenomenon, we propose Chain of Merges (CoM), a novel approach that updates activation statistics autoregressively, capturing inter-layer dependencies and fully eliminating MCS throughout the network. Empirical results on standard vision and language benchmarks demonstrate that CoM consistently outperforms existing methods across diverse architectures and domains. Looking ahead, we plan to extend the proposed framework to a wider range of activation-based merging techniques and to include experiments on Federated Learning scenarios, where efficiently merging local clients' updates is essential for centralized performance.

**Reproducibility Statement.** We include our codebase in the supplementary material, which requires the dataset to be available locally on disk.

**Use of Large Language Models.** For this work, we used LLM-based tools only to improve the grammar, spelling, and phrasing of text that we had already written. No LLM was used to generate research ideas, derive algorithms, design experiments, analyze results, or write any sections of the paper from scratch. All technical content (problem formulation, method design, implementation, experiments, and interpretation of results) is entirely our own, and we take full responsibility for it.

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

# APPENDIX

## A  DERIVATION OF THE SIMILARITY-WEIGHTED MERGING SOLUTION

We want to minimize the following objective with respect to the matrix $\boldsymbol{W}_M^l$:

$$\tilde{\Omega}_l = \sum_{i=1}^{N} \omega_i^l \, \|\boldsymbol{W}_M^l \tilde{\boldsymbol{X}}_i^l - \boldsymbol{W}_i^l \tilde{\boldsymbol{X}}_i^l\|_2^2.$$

To simplify the notation, define $\boldsymbol{G}_i^l = \tilde{\boldsymbol{X}}_i^l (\tilde{\boldsymbol{X}}_i^l)^\top$. Expanding the norm and applying standard rules of matrix differentiation, the gradient of the objective with respect to $\boldsymbol{W}_M^l$ is:

$$\nabla_{\boldsymbol{W}_M^l} \tilde{\Omega}_l = 2 \left[ \boldsymbol{W}_M^l \Big( \sum_{i=1}^{N} \omega_i^l \boldsymbol{G}_i^l \Big) - \Big( \sum_{i=1}^{N} \omega_i^l \boldsymbol{W}_i^l \boldsymbol{G}_i^l \Big) \right].$$

The minimizer is obtained by setting the derivative equal to zero as:

$$\boldsymbol{W}_M^l \Big( \sum_{i=1}^{N} \omega_i^l \boldsymbol{G}_i^l \Big) = \sum_{i=1}^{N} \omega_i^l \boldsymbol{W}_i^l \boldsymbol{G}_i^l.$$

Finally, we can solve for $\boldsymbol{W}_M^l$ by multiplying on the right hand side with the inverse of $\sum_{i=1}^{N} \omega_i^l \boldsymbol{G}_i^l$:

$$\boldsymbol{W}_M^l = \left( \sum_{i=1}^{N} \omega_i^l \boldsymbol{W}_i^l \boldsymbol{G}_i^l \right) \left( \sum_{i=1}^{N} \omega_i^l \boldsymbol{G}_i^l \right)^{-1}.$$

If the latter matrix is singular, the Moore–Penrose pseudoinverse should be used instead.

## B  SUPPLEMENTARY ABLATIONS FOR TEXTUAL DATASETS

**Similarity-based importance.**    In Figure 4, we report the similarity-based importance analysis for textual tasks (SNLI, MNLI, SICK, QNLI, RTE, and SciTail). Following the approach used for vision datasets, we compute the performance gap between task-specific fine-tuned models and the zero-shot performance of the base Llama3-8B model, and compare it to the similarity-based weighting factor $\omega_i^l$ averaged across layers.

In contrast to vision tasks, textual datasets exhibit a relatively uniform importance distribution: both performance gaps and similarity weights vary little across tasks. As a result, the correlation between the two measures is weaker, and similarity-weighted merging provides limited benefit. These findings highlight that the effectiveness of similarity-based weighting depends on task heterogeneity: greater diversity relative to the pretraining distribution amplifies its impact on the merged model.

**Measuring MCS.**    We extend the analysis from Section 5 to Llama3-8B, computing MCS for each layer using the same methodology. As illustrated in Figure 3, textual datasets display merging covariate shifts that are broadly comparable to those observed in vision models.

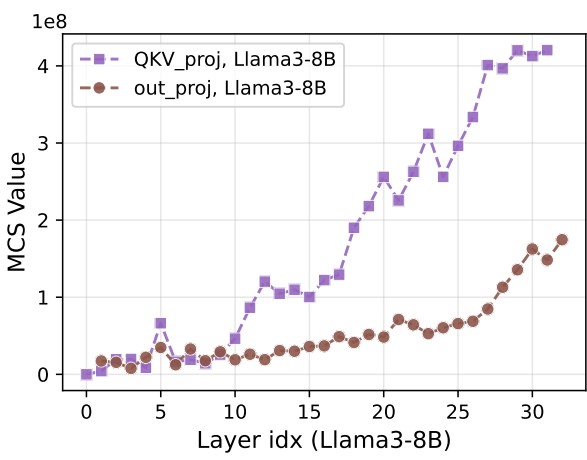

Figure 3: Merging covariate shift across layers.

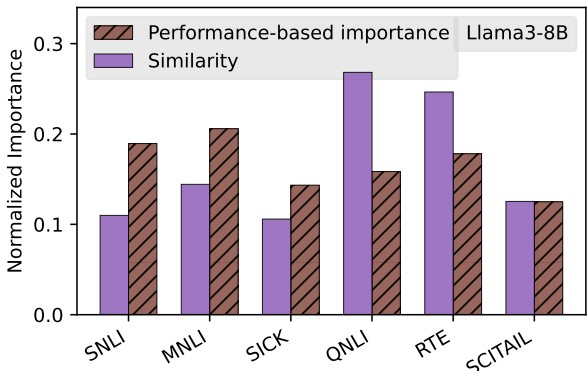

Figure 4: Task importance *vs.* similarity.

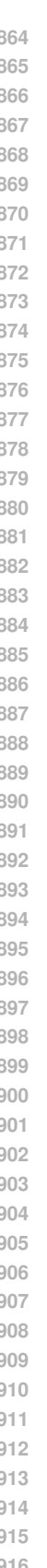

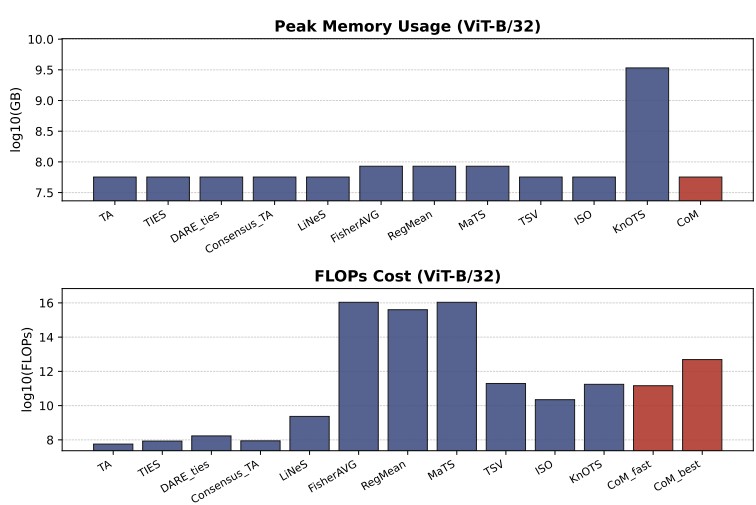

(a) Peak memory usage and merging computational cost for Vit-B/32.

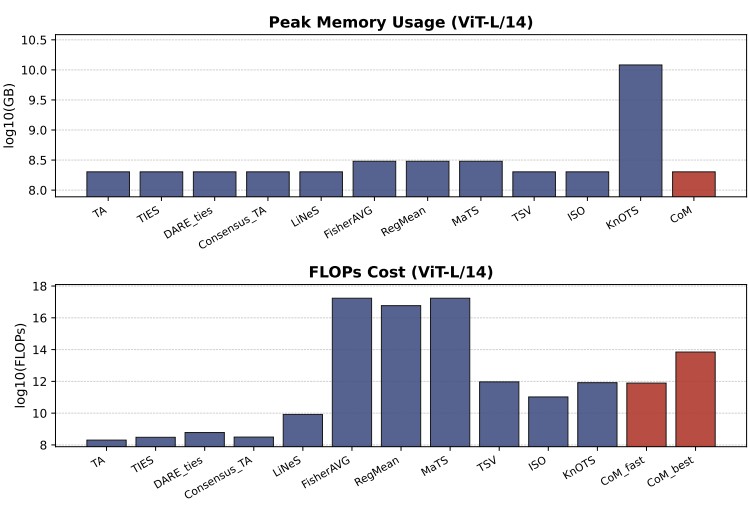

(b) Peak memory usage and merging computational cost for ViT-L/14.

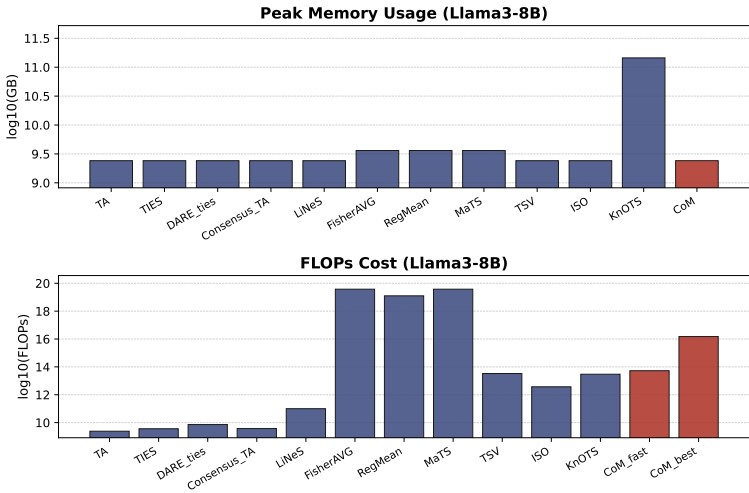

(c) Peak memory usage and merging computational cost for Llama3-8B.

Figure 5: Comparison of merging methods across computational and memory cost.

## C   COMPUTATIONAL AND MEMORY COST

We evaluate the efficiency of the proposed methodology in terms of computational complexity and memory usage, comparing it against established techniques. For the three architectures used, Figures 5a to 5c (log-scale) report the number of FLOPs as a measure of computational cost, and the theoretical memory overhead in GB. We denote our methodology with the minimum number of samples achieving state-of-the-art performance as `CoM_fast`, and the variant using the optimal number of samples for maximum performance as `CoM_best`. The results show that CoM achieves performance comparable to alternative merging strategies. For clarity, methods such as Task Arithmetic, TIES, DARE$_{\text{TIES}}$, Consensus TA, and LiNeS are omitted from the computational plot, since their complexity is negligible.

A notable comparison is with RegMean. Although our approach shares a similar formulation, it differs in the update rule by using the merged activations instead of the task-specific ones. This key distinction improves accuracy while keeping the number of forward passes unchanged: RegMean requires one forward pass per task-specific model to compute the Gram matrices, whereas CoM performs the same number of passes on the merged model. Finally, CoM requires fewer examples in practice, resulting in consistently lower computational cost, as shown in Table 4.

These findings highlight that the proposed recursive scheme provides an effective balance between efficiency and performance, ensuring state-of-the-art accuracy while keeping both computation and memory overhead limited.

## D   OUT-OF-DISTRIBUTION GENERALIZATION

**Experimental protocol.**   To assess the generalization capabilities of our proposed CoM method, we evaluate its transfer performance on tasks held out during the merging process, following the protocol introduced in AdaMerging. Specifically, we utilize the standard 8-Vision benchmark and define two distinct Out-of-Distribution (OOD) evaluation splits. For consistency and reproducibility, we employed the same fully fine-tuned checkpoints used by AdaMerging and mirrored their tasks split.

In each split, models are fine-tuned on six datasets, designated as In-Distribution (ID) tasks. specifically, Split 1 involves merging models finetuned on SUN397, Cars, RESISC45, DTD, SVHN and GT-SRB, leaving out MNIST and EuroSAT as the OOD tasks. Split 2 instead, involves merging models finetuned on SUN397, Cars, GTSRB, EuroSAT, DTD and MNIST, leaving out RESISC45 and SVHN. We then merge these models and assess the resulting merged model's performance on the two remaining datasets, which constitute the OOD tasks. This design directly mirrors the OOD setup of AdaMerging, effectively isolating and measuring the merging method's ability to generalize beyond the data domains on which it was optimized.

Table 5: Absolute average accuracies (%) of merged models with ViT-B/32 in Out-Of-Distribution setting using 2 different tasks splits. Best results in bold, second-best underlined.

| | Split 1 | | Split 2 | |
|---|---|---|---|---|
| **Method** | **ID** | **OOD** | **ID** | **OOD** |
| TA | 70.6 | 61.7 | 73.9 | 51.1 |
| TIES | 69.3 | 59.6 | 71.8 | 53.9 |
| Consensus TA | 72.8 | 59.5 | 73.9 | 53.3 |
| LiNes | 73.6 | 59.2 | 75.9 | 52.3 |
| AdaMerging | 77.4 | **70.0** | 80.3 | 55.5 |
| AdaMerging++ | 78.0 | 68.7 | 80.8 | **58.5** |
| TSV | 76.8 | 59.3 | 79.0 | 51.9 |
| **CoM (ours)** | **81.1** | 64.5 | **83.4** | 56.2 |

**Results.**   As shown in Table 5, CoM demonstrates strong robustness across both OOD splits. For ID tasks, it achieves state-of-the-art performance in all scenarios, consistently surpassing all baselines and aligning with the trends presented in our main experiments (Section 4). For OOD tasks, AdaMerging remains the top-performing approach. This is predictable, as AdaMerging is the sole method employing post-hoc training to reduce entropy and optimize knowledge transfer to unseen tasks. Crucially, among all methods that forgo post-hoc optimization, CoM is the strongest performer, consistently ranking immediately behind AdaMerging and outperforming all other approaches.

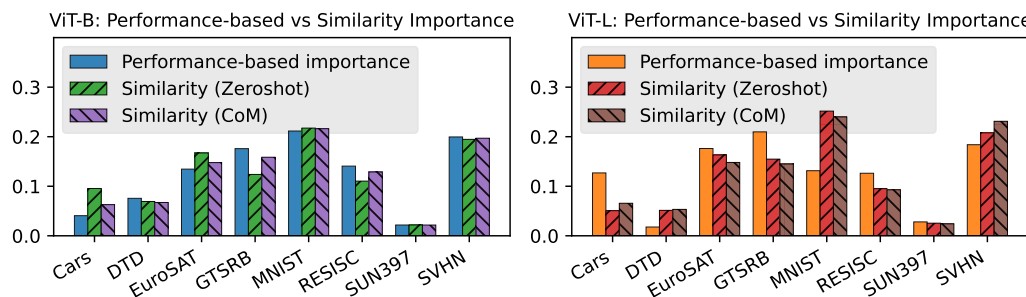

Figure 6: **Ablation of Task Importance Coefficients (ViT-B and ViT-L).** We compare the importance scores derived from our proposed CoM method (which uses the merged model as a reference) against those derived from the Zero-shot pretrained model and a performance-based oracle. The results demonstrate that CoM coefficients strongly correlate with the Zero-shot reference, justifying the use of the merged model as an efficient proxy.

# E  ANALYSIS OF TASK SIMILARITY COEFFICIENTS

In this section, we provide a deeper analysis of the task similarity coefficients used in our CoM method. We first elaborate on the theoretical connection between feature correlation and generalization, and subsequently validate our choice of using the merged model as a reference for computing these statistics.

## E.1  THEORETICAL MOTIVATION: ORTHOGONALITY AND GENERALIZATION

Our approach quantifies inter-feature correlation to estimate task similarity. We argue that for in-distribution inputs, representations from large-scale pretrained models are approximately decorrelated, as they capture broad, general-purpose structures rather than task-specific patterns.

Consequently, high off-diagonal correlations in the input Gram matrix $G$ indicate that a specific task concentrates on a narrow subspace of the original data distribution, diverging significantly from the pretraining initialization. Intuitively, when a task relies heavily on a limited set of feature directions, those features exhibit higher correlation, revealing that the model is focusing on a restricted region of the representation space. This perspective aligns with established literature demonstrating that feature decorrelation is linked to improved generalization (Cogswell et al., 2015; Morcos et al., 2018). Our method leverages this principle: we treat significant deviations from the decorrelated pretrained state (high inter-feature correlation) as a signal of task specificity.

## E.2  ABLATION ON REFERENCE MODELS

To maintain comparability across tasks and layers, similarity coefficients must be computed using a single, task-agnostic reference model. Using task-specific fine-tuned models as references would be improper, as this would quantify how well a model fits its own specific task rather than providing a standardized measure of distributional shift.

Ideally, the zero-shot (pretrained) model serves as the ground truth reference. However, for computational efficiency within our pipeline, CoM computes correlations using the merged model. We posit that the merged model is a valid proxy because it remains approximately task-agnostic and provides a balanced representation across all tasks.

To validate this approximation, we conducted an ablation study comparing the task importance scores derived from three different sources:

1. **Performance-based importance (Oracle):** Importance scores calculated based on actual evaluation performance.

2. **Similarity (Zero-shot):** Coefficients computed using the original pretrained checkpoint (the ideal reference).

3. **Similarity (CoM):** Coefficients computed using our proposed method with the merged model (the efficient proxy).

The results, illustrated in Figure 6 for both ViT-B and ViT-L architectures, show a high degree of alignment between the coefficients computed via CoM (purple/brown hatched bars) and those computed via the Zero-shot model (green/red hatched bars). This confirms that using the merged model to compute the Gram matrices does not introduce significant deviation from the ideal pretrained reference, validating the efficiency of our protocol. Furthermore, both similarity-based metrics track the general trends of the performance-based oracle, particularly in identifying high-importance tasks such as EuroSAT and SVHN.

