# OpenReview forum: "Rethinking Layer-wise Model Merging through Chain of Merges"
_ICLR.cc/2026/Conference — Submitted to ICLR 2026_

### Official Review · Reviewer_SXdo · 2025-11-01

**Soundness:** 1
**Presentation:** 2
**Contribution:** 2
**Rating:** 2
**Confidence:** 4

**Summary:**

The paper introduces Chain of Merges (CoM), a new layer-wise model merging technique. The authors identify "merging covariance shift" (MCS) as a key problem in which merging layers independently induces activation shifts that negatively affect subsequent layers. CoM addresses this problem by merging layers sequentially, while updating activation statistics to ensure consistency.

**Strengths:**

- Model merging is a critical and practical research area for making AI systems more modular.
- The identification of MCS is an interesting conceptual contribution.
- The paper is well-written and clear.
- The empirical results, within the context of LoRA-finetuned models, appear strong.

**Weaknesses:**

- **Misleading scope**. The paper's entire experimental validation is performed exclusively on LoRA-finetuned models. This fundamental design choice is only mentioned deep in the "Implementation Details" (Section 4.1). The abstract, introduction, and methodology sections all speak in general terms of merging "fine-tuned models" and "task-specific checkpoints," which strongly implies full fine-tuning. The paper is therefore not evaluated as a general model-merging technique but as a LoRA-merging technique, and its contributions must be re-evaluated in this much narrower context.
- **Invalid SOTA comparisons**. This LoRA-only setup invalidates the claims of "state-of-the-art" performance in model merging. The majority of the cited literature and the baselines themselves were developed and evaluated in the context of full-parameter fine-tuning. A clear example is TSV: the authors report ~78% normalized accuracy (on ViT-L/14), whereas the original TSV paper reports ~97% on the same benchmark (likely starting from an even higher absolute accuracy – which is not reported).
- **No absolute accuracy**. The paper indeed reports only normalized accuracy. The omission of absolute accuracy is a critical flaw. Normalized accuracy alone can be misleading, as it may mask poor overall performance. Given the baseline discrepancies, reporting also absolute accuracy is essential for a fair comparison.

**Questions:**

- What is the performance of CoM on models finetuned without adapters?
- Can the authors provide the absolute accuracies (not just normalized) for all experiments?

---

> ### Author Response · Authors · 2025-11-24
>
> > Misleading scope. The paper's entire experimental validation is performed exclusively on LoRA-finetuned models.
>
> > Invalid SOTA comparisons. This LoRA-only setup invalidates the claims of "state-of-the-art" performance in model merging.
>
> Thank you for raising this helpful concern. We believe it allowed us to enhance our work and provide a better evaluation.
>
> Our initial focus on **LoRA** checkpoints followed common practice in recent merging work (e.g., Stoica et al., ICLR 2025; Panariello et al., NeurIPS 2025), and we tried to make that scope clear in the manuscript. That being said, we agree broader validation is important. To enhance generality and comparability, we have now **added full-parameter fine-tuning (FFT)** experiments. These new results corroborate CoM’s effectiveness and have been **integrated into the revised manuscript**. Also with FFT checekpoints, CoM demonstrate its effectiveness (SOTA performance) on all settings.
>
> |**Method**| **Norm (B/32)**|**Abs (B/32)**|**Norm (L/14)**|**Abs (L/14)**|
> |-|:-:|:-:|:-:|:-:|
> TA | 76.5 | 70.8 | 88.7 | 84.9 |
> TIES | 81.2 | 75.1 | 90.8 | 86.9 |
> Consensus TA | 81.4 | 75.0 | 90.2 | 86.3 |
> FisherAVG | 73.8 | 68.3 | 85.9 | 82.2 |
> RegMean | 77.6 | 71.8 | 87.5 | 83.7 |
> Loc-and-Stitch | 86.3 | 79.9 | 90.4 | 86.5 |
> AdaMerging++ | 87.6 | 81.1 | 95.1 | 91.0 |
> ProDistill | 92.9 | 86.0 | 96.1 | 91.9 |
> TSV | 92.8 | 85.9 | 97.2 | 93.0 |
> **CoM (ours)**   | **94.8** | **87.7** | **97.8** | **93.6** |
>
> > No absolute accuracy. The paper indeed reports only normalized accuracy.
>
> We have included the absolute accuracy for all experiments in the main paper. With this modification, our conclusions and claims remain unchanged, as our method achieves state-of-the-art results in both cases.
>
> We remain available to address any additional questions or concern.

---

### Official Review · Reviewer_Pa3c · 2025-11-01

**Soundness:** 3
**Presentation:** 3
**Contribution:** 3
**Rating:** 4
**Confidence:** 4

**Summary:**

The paper introduces a new methodology for model merging. In particular, the paper notes that previous layer-wise merging works have neglected the well-known internal covariate shift problem, resulting in a distributional mistmatches. As such, the paper proposes a Chain of Merges (CoM) method, which sequentially merges weights layer by layer. Experimental results demonstrate that the proposed method provides very high normalized accuracy.

**Strengths:**

- The proposed idea is well-motivated, based on the internal covariate shift problem.
- The displayed experimental results show strong performance of the proposed method.
- The paper is well-written and easy to follow.

**Weaknesses:**

- The paper does not explicitly explain the number of examples used in the method. One may guess that the number of examples is 500 based on the ablation study results in Table 4, but the performance of ViT-B/32 and Llama3-8B do not match with the main results in Table 1 and Table 2.
- Also, the paper does not specify which set the examples come from.
- The paper only reports the normalized accuracy, without reporting the actual performance. Considering how the proposed method is applied on LoRA-fine-tuned models, the results as is do not give a full picture as to how the proposed method actually performs in comparison on other works.
- The paper does not show large-scale experiments (14 tasks or 20 tasks, as done so in Consensus TA (Wang et al., 2024)).
- The paper does not compare against more recent state-of-the-art methods, such as EMR-Merging [A].
- Some works compared in the table originally reported the results with full-fine-tuned models, to the best of the reviewer's knowledge. The comparison does not seem completely fair.


[A] Huang et al., Emr-merging: Tuning-free high-performance model merging. NeurIPS 2024.

**Questions:**

- Why is the proposed method only applied on LoRA-fine-tuned models? How is the performance when applied to full-fine-tuned models?
- What is the actual number of examples used?
- Do examples come from training set or test set?
- The memory overhead seems to be very similar to TA. This is a bit confusing and surprising. If the Gram matrix is computed on at least 500 samples, wouldn't there be more memory usage? What's the actual memory usage?
- How is the computational cost and latency compared to previous works mentioned in Table 1?

---

> ### Author Response · Authors · 2025-11-24
>
> **Number and source of examples.** $~~$ In our experiments, we use backbone-specific budgets:
>
> - **ViT-B/32:** 1,000 examples
> - **ViT-L/14:** 500 examples
> - **Llama-3-8B:** 250 examples
>
> Rationale: the effective sample size in our Gram statistics scales with the **sequence length** (tokens/patches) that flows along the batch axis; shorter sequences require **more examples** to obtain stable estimates (*e.g.*, ViT-B/32), whereas longer sequences need fewer. Table 4 employs a fixed budgets for the ablation (to isolate sample efficiency), while Tables 1–2 report backbone-specific budgets as above. We will add these exact counts and splits to the manuscript for clarity.
>
> Examples are drawn from the **validation** set (the same split that other baselines use for their hyperparameter search). Using the training set produces similar accuracy. We **never** use test examples.
>
> **Absolute performance.** $~~$ Normalized scores are a fixed rescaling of absolute accuracies (by the mean accuracy of the fine-tuned models), so rankings and margins are preserved. We will add per-task **absolute accuracies** to the manuscript for a better comparison.
>
> **LoRA vs. full-rank.** $~~$ Consistent with recent model-merging works (Stoica et al., ICLR 2025; Panariello et al., NeurIPS 2025), our primary experiments leverage **LoRA** checkpoints. We now report a **full-rank fine-tuning evaluation** in the revised version of our manuscript, and report them below for completeness. These additional results confirm CoM’s effectiveness, yielding state-of-the-art performance (with a smaller margin, in this case).
>
> |**Method**| **Norm (B/32)**|**Abs (B/32)**|**Norm (L/14)**|**Abs (L/14)**|
> |-|-|-|-|-|
> TA | 76.5 | 70.8 | 88.7 | 84.9 |
> TIES | 81.2 | 75.1 | 90.8 | 86.9 |
> Consensus TA | 81.4 | 75.0 | 90.2 | 86.3 |
> FisherAVG | 73.8 | 68.3 | 85.9 | 82.2 |
> RegMean | 77.6 | 71.8 | 87.5 | 83.7 |
> Loc-and-Stitch | 86.3 | 79.9 | 90.4 | 86.5 |
> AdaMerging++ | 87.6 | 81.1 | 95.1 | 91.0 |
> ProDistill | 92.9 | 86.0 | 96.1 | 91.9 |
> TSV | 92.8 | 85.9 | 97.2 | 93.0 |
> **CoM (ours)**   | **94.8** | **87.7** | **97.8** | **93.6** |
>
> **Comparison with EMR-Merging**
> EMR-merging augments the merged weights with task-specific modulators (a mask and a rescaler) that are applied at inference time. This means that the effective parameters depend on the input/task at test time: *i.e.*, **EMR-merging employs N different models, one for each task**. By contrast, our setting follows the standard "static" model-merging protocol: a single set of parameters is produced and used for all test inputs: no input-dependent routing, masks, or adapters is allowed at inference. A direct, apples-to-apples comparison is therefore out of scope for our protocol, as EMR-merging **does not comply** with our experimental setting.
>
> - Yang et al., 2024. Model merging in llms, mllms, and beyond: Methods, theories, applications and opportunities.
>
> - Wang et al., (2025). Scaling Intelligence Through Model Merging: A Comprehensive Survey.
>
> **Additional experiments on more tasks.** We appreciate the suggestion; expanding to a broader set of tasks is beyond our current possibilities due to the time constraints of the rebuttal period, but will be added on our roadmap for follow-up works.
>
> > The memory overhead seems to be very similar to TA. This is a bit confusing and surprising.
>
> **Computational footprint.** $~$ CoM processes **one layer at a time** and, within that layer, **one batch at a time**. For each active layer (hidden size $d$), it accumulates a single **feature–feature Gram** of size $d \times d$ that **does not grow with the number of samples**. Adding more samples only increases wall-clock time (via more batches), not memory. Consequently, the overhead beyond model weights is small, and CoM’s footprint remains comparable (on a log scale, Figure 5) to simpler baselines.
>
>
> > How is the computational cost and latency compared to previous works mentioned in Table 1?
>
> **Computational cost.** $~~$ We report computational cost (FLOPs) in Figure 5 (Appendix). The computational cost of CoM is higher than ISO, comparable to KnOTS and TSV when using few examples, and lower than RegMean, FisherAVG and MaTS in any case.
>
> ### **References**:
>
> - Panariello et al., Accurate and Efficient Low-Rank Model Merging in Core Space. In NeurIPS, 2025.
>
> - Stoica et al., Model merging with SVD to tie the Knots. In ICLR, 2025.

---

> > ### Comment · Reviewer_Pa3c · 2025-11-27
> >
> > While the reviewer appreciates the rebuttal, the concerns remain, especially regarding the comparison against EMR-Merging.
> > The reviewer disagrees with the authors on the fairness of comparison. The proposed method uses 500-1000 examples for ViT models, while EMR-Merging does not. With the author's argument, it is not fair to compare the proposed method against other methods, such as TSV, that do not use examples. Furthermore, EMR-Merging proposes a practically efficient approach, that stores bit-wise masks, instead of parameters of N different models.
> > Lastly, if the fairness of comparisons is considered, the current experimental results do not seem to provide extensive comparisons against other works that employ examples [B, C].
> >
> > [B] Lu et al., Twin-merging: Dynamic integration of modular expertise in model merging. NeurIPS 2024.
> > [C] Tang et al., Merging multi-task models via weight-ensembling mixture of experts. ICML 2024.

---

> ### Author Response · Authors · 2025-11-27
>
> We sincerely thank the reviewer for their follow-up. Below, we address their points in more detail, and we have updated our manuscript by incorporating:
> - details on the static merging vs. dynamic merging experimental protocol;
> - a discussion on the suggested methodologies, citing them in Section 6.
>
> ### 1. Experimental **setting** vs. EMR-Merging and related methods
>
> Our work, like **all evaluated baselines in our experiments** is designed around what we view as the **standard static model-merging protocol**, as adopted in recent surveys (e.g., Yang et al., 2024; Wang et al., 2025):
>
> - We produce **a single merged set of parameters**.
> - At inference time, **all tasks and all inputs** are processed using this **same set of weights**.
> - The protocol **does not allow task IDs, task-dependent routing**, or any input-conditioned modification of the merged parameters at test time.
>
> By contrast, the methods you highlight $-$ **EMR-Merging, Twin-Merging, and WeMoE** $-$ rely on conditional inference, in which the effective **parameters depend on the task or input at test time**:
>
> - **EMR-Merging (Huang et al., NeurIPS 2024)** augments the merged model with task-specific masks and rescaling factors. Even though these are stored efficiently (bit-wise), the inference behavior remains **task-conditional**, since different tasks activate different modulators.
> - **Twin-Merging (Lu et al., NeurIPS 2024)** employs dynamic integration of modular expertise. Its modules are weighted depending on a separate router model, which uses the input to choose the task.
> - WeMoE (**Tang et al., ICML 2024**) also relies on a weight-ensembling mixture of experts, again yielding **input- or task-dependent** effective parameters at inference chosen by a separate router model.
>
> We fully agree that these approaches are important and technically impressive contributions to the model-merging landscape. Our concern is not with their quality, but rather with the **difference in evaluation protocol**:
>
> - In our setting, there is **exactly one parameter vector** after merging, and no routing mechanism. We store one single merged parameter vector, used for all examples and tasks.
> - In those methods, even when storage is compressed, the inference-time model behaves more like a **conditional ensemble or MoE system** than a single static model.
>
> For this reason, we initially judged that a direct numerical comparison could be misleading, since it would be juxtaposing **different deployment regimes**. We understand this point might not have been sufficiently clear in the current manuscript, and we are grateful that your comment gives us the opportunity to make this distinction explicit.
>
> ---
>
> ### 2. Use of examples and fairness to example-free baselines (e.g., TSV)
>
> We fully acknowledge your concern that CoM and other baselines use examples during merging (e.g., FisherAVG, RegMean, AdaMerging, ProDistill), whereas some others (e.g., TA, TIES, Consensus, TSV) do not.
>
> Specifically:
> - FisherAVG leverages data to estimate the Fisher Information Matrix;
> - AdaMerging and ProDistill perform gradient descent to optimize layer-wise coefficients;
> - RegMean uses input grams (as our method) for its closed-form merging solution.
>
> However, **all methodologies in the table** also heavily rely on a hyperparameter search on the validation set, and hence they **require the access to the validation data**.
>
> From this perspective, we believe that comparing CoM to **static, example-free baselines** like TSV is still **meaningful**, as CoM leverages (a subset of) the same examples that are already used for hyperaparameter selection. Instead, comparing CoM directly to **conditional, multi-expert or routed methods** like EMR-Merging, Twin-Merging or WeMoE is more akin to comparing a single static model to a (compressed) multi-expert system, which we view as a conceptually different design point: it includes presence of a router model or task oracle, more forward passes, and model parameters are modified for each examples.
>
> Thanks to your comments, we have updated the manuscript to more transparently reflect our assumptions and limitations. In particular, we have:
>
> 1. **Explicitly formalized our evaluation protocol**, clearly stating that we follow a "single static merged model, no conditional routing, no task IDs at inference" protocol.
> 2. **Add a dedicated subsection on static vs. conditional merging**, carefully explain why methods such as EMR-Merging, Twin-Merging, and Tang et al. are highly relevant but operate under a different inference regime.
>
> Once again, we are very grateful for your detailed and constructive feedback. Your comments have helped us sharpen the conceptual framing of our work and improve the transparency of our experimental setting and comparisons.
>
> - Yang et al., 2024. Model merging in llms, mllms, and beyond: Methods, theories, applications and opportunities.
>
> - Wang et al., (2025). Scaling Intelligence Through Model Merging: A Comprehensive Survey.

---

> > ### Comment · Reviewer_Pa3c · 2025-11-28
> >
> > The reviewer thanks the author for the follow-up.
> > But, the reviewer disagrees on the statement that the comparison against TSV is fair, where the authors argue that both TSV and the proposed method CoM use the same validation set.
> > TSV does not use the validation set to tune the hyperparameters. The paper of TSV states using the $\alpha=1.0$ without hyperparameter tuning, as also emphasized in Table 1 of the paper.
> > Furthermore, the reviewer does not understand why the comparison should be exclusive, based on whether the comparison is inclusive or exclusive, when, in the end, the objective is the same.
> > If the authors are concerned about the multiple passes (although EMR-Merging does not) or other computational complexities, the comparisons set-up could be more convincing if there is comprehensive analysis on the memory and computational complexities of previous works and the proposed method.

---

> > > ### Author Response · Authors · 2025-12-02
> > >
> > > Thank you for the clarification. To avoid any misunderstanding, our point was simply that several baselines in the main tables do rely on data for merging (e.g., RegMean, FisherAVG, MaTS, AdaMerging, ProDistill), and these methods are commonly regarded as **fair comparisons in the literature (including in the TSV paper)**. Moreover, all baselines perform a grid search to select the best $\alpha$; TSV included.
> > >
> > > We agree with the reviewer that TSV demonstrates, through an ablation study, that its performance is relatively insensitive to the choice of $\alpha$. On our side, we emphasize that CoM uses a fixed $\alpha=1$, with **no hyperparameter tuning or dataset-specific adjustment**.
> > >
> > > Finally, following your suggestion, we include a comprehensive computational and memory footprint analysis in the following, covering CoM and the dynamic merging methods you mentioned.
> > >
> > > # Quantitative Footprint of CoM vs. Dynamic Merging Methods
> > >
> > > With the following tables we want to clarify that **CoM belongs to the branch of "single-model/static-merging" approaches**, whereas EMR-Merging, Twin-Merging, and WEMoE belong to **conditional, multi-expert regimes**. Indeed, as shown in the tables below, CoM uses exactly the same memory footprint as Task Arithmetic **during inference**: a single model and nothing more; no auxiliary masks, no routing networks, and no task-dependent storage. For the sake of fairness, we compared only with methods that follow this same design.
> > >
> > > ## **ViT-B/32, 8Vision (87.1M parameters)**
> > >
> > > |Method|Param multiplier ($\times$ num. of parameters)|Total params (M)| memory (fp16, MB)|
> > > |-|-|-|-|
> > > |Task Arithmetic|1.0$\times$|87.1|166|
> > > |CoM|1.0$\times$|87.1|166|
> > > |Twin-Merging (router + compressed experts)|1.8$\times$ (c=0.1)|156.8|299|
> > > |EMR-Merging (static+task-conditional) | 9.0$\times$ (1 model + 8 binary masks)|783.9|249|
> > > |WEMoE (conditional MoE) |6.2 $\times$ (1 model + 8 "half models": i.e., MLP layers only)|470.3|897|
> > >
> > > ## **ViT-L/14, 8Vision (304.2M parameters)**
> > >
> > > |Method|Param multiplier($\times$ num. of parameters)|Total params (M)|memory (fp16, MB)|
> > > |-|-|-|-|
> > > |Task Arithmetic| 1.0$\times$|304.2|580|
> > > |CoM|1.0$\times$|304.2|580|
> > > |Twin-Merging|1.8$\times$  (c=0.1)| 547.6|1509|
> > > |EMR-Merging|9.0$\times$ (1 model + 8 binary masks)|2737.8|725|
> > > |WEMoE|6.2$\times$ (1 model + 8 "half models": i.e., MLP layers only)|1642.7|3133|
> > >
> > > **Twin-Merging** is extremely compact, but remains a **conditional** method that uses an additional **router network** at inference. Hence, the system is **not** a single static model $-$ its actual parameters are personalized to each test input.
> > >
> > > **EMR-Merging requires storing eight full binary masks (one per task)**. These are compact, but still add **+83 MB** of additional memory for ViT-B/32 and **+145 MB** for ViT-L/14. This yields a **1.5$\times$ increase in fp16 memory** relative to a single model. Most importantly, EMR-Merging $-$ like Twin-Merging $-$ is a conditional approach, and it **requires a task label for each example** at inference time.
> > >
> > > WEMoE stores multiple MLP experts for each task, activating each expert during the forward pass depending on the input. For transformers models, where MLPs represent $\approx$ 65% of parameters, this results in **6.2$\times$** more parameters.
> > >
> > > ---
> > >
> > > In the following, we additionally report the computational footprint (evaluated at test time) for the aforementioned approaches.
> > >
> > > **Summary.** Our approach does not incur any additional cost during inference, whereas conditional systems incur **substantially higher inference complexity** due to task- or input-dependent expert evaluation.
> > >
> > > ### **ViT-B/32, 8Vision ($\approx$ 4.41 GFLOPs)**
> > >
> > > |Method|Factor vs single|FLOPs/image (GFLOPs) |
> > > |-|-|-|
> > > |Task Arithmetic|1.0$\times$|4.41|
> > > |CoM|1.0 $\times$|4.41|
> > > |EMR-Merging|$\approx$ 1.0 $\times$|4.41|
> > > |Twin-Merging|$\approx$ 2.4 $\times$|10.71|
> > > |WEMoE (1 shared + 8 task MLPs)|$\approx$ 6.2 $\times$|27.34|
> > >
> > > ---
> > >
> > > ### **ViT-L/14, 8Vision (≈81.0 GFLOPs)**
> > >
> > > | Method| Factor vs single | FLOPs / image (GFLOPs) |
> > > |-|-|-|
> > > |Task Arithmetic|1.0$\times$|81.0|
> > > |CoM|1.0$\times$|81.0|
> > > |EMR-Merging|$\approx$ 1.0$\times$|81.0|
> > > |Twin-Merging| $\approx$ 1.9$\times$|155.5|
> > > |WEMoE (1 shared + 8 task MLPs)|$\approx$ 6.2$\times$|502.2|
> > >
> > > The tables show that, since **CoM and Task Arithmetic** evaluate a **single unified parameter vector**, they retain the exact computational complexity of a single ViT backbone. EMR-Merging, unlike other dynamic methods, avoids using an expert router module by **requiring the test ground-truth task label directly**. By so doing, it introduces only a *minor operational overhead* due to handling its bit-wise masks, which does not affect the overall FLOPs. **Twin-Merging** requires nearly double the compute (≈1.9-2.4×) as it mandates the parallel execution of both shared and exclusive experts for every input. Finally, **WEMoE** drastically increases complexity to 6.2× by evaluating eight additional experts per block.

---

### Official Review · Reviewer_1dpG · 2025-11-01

**Soundness:** 3
**Presentation:** 2
**Contribution:** 2
**Rating:** 4
**Confidence:** 4

**Summary:**

This paper proposes CoM, which recursively merges the linear layer weights from first layer to last layer during model merging. It improves Regmean by addressing internal covariate shift and demonstrates good empirical performance.

**Strengths:**

1. The paper is clearly-written and easy to understand.
2. The method demonstrates good empirical results in both vision and NLP, including the results obtained with Llama-3.

**Weaknesses:**

1. The proposed method CoM offers modest novelty. It extends Regmean by fixing the internal covariate shift problem. The novelty primarily lies in providing the inputs of the merged model rather than the individual models for Regmean merging algorithm.

2. The performance of the baseline methods seem concerningly low (Table 1). The performance of most merging-based methods, including TA [1], Ties [2], DARE [3] seem to be notably lower than their reported results in the original papers. Furthermore, the methods which are designed to improve over TA, such as TIES, DARE, Consensus [4], LiNeS [5], underperform or have the same performance as TA. Furthermore, the manuscript does not provide implementation details to the compared baselines, which are needed in this case as the the baselines do not perform as expected.

Note: If the reason for the baseline methods’ low performance is due to applying LoRA (which case also needs to be clarified), I suggest the authors add results for standard fine-tuning as well, following the literatures in model merging [1-5]. It is important to evaluate the performance of the proposed method in the standard fine-tuning scenario as well.

3. The computational cost of TA, TIES, DARE, Consensus and LiNeS are missing in the computational cost comparison. It is unfair to omit them simply because their computational cost is negligible (Section C). The computational cost of the proposed methods need to be compared to the light-weight methods as well to clearly demonstrate the tradeoff to the viewers.

4. Many important experimental details are not given.

4.1 The implementation details of the baselines, as stated before.

4.2 Where did the authors get the input samples for calculating the gram matrix, i.e., are they the same or different samples in the test set?

4.3 How is the method merging the layers that are not linear layers (e.g., the attention layers)? Does it follow the Regmean setup to average their weights?

**Questions:**

1. For Eq. 5, even for the first linear layer, there are attention layers before the linear layers in the same residual block. As a result, shouldn’t the input still be affected by prior merging (on the attention layers in the same block, whose merging process is not explained in the paper) as well, making the notations in Eq. 5 inaccurate?

[1] Editing Models with Task Arithmetic
[2] TIES-Merging: Resolving Interference When Merging Models
[3] Language Models are Super Mario: Absorbing Abilities from Homologous Models as a Free Lunch
[4] Localizing Task Information for Improved Model Merging and Compression
[5] LiNeS: Post-training Layer Scaling Prevents Forgetting and Enhances Model Merging

---

> ### Author Response · Authors · 2025-11-24
>
> ### 1. **Novelty**
> The central contribution of our proposal is to **identify, formalize, and fix merging covariate shift (MCS)**: a shift between the activations of the merged network and those of the original fine-tuned models. Prior layer-wise merging methods leveraging input data (*e.g.*, RegMean, MaTS, ProDistill, ZipIT) implicitly assume stationary activations. Therefore, they regress on the original layer inputs, which are **mismatched once** parameters are **merged**. Instead, we identify and formalize MCS, proposing a novel merging technique that performs regression on merged-model activations, refreshing them autoregressively after each layer to ensure distributional consistency throughout the network. We believe this is a substantive contribution, which we prove it has a tangible practical impact, because explicitly modeling and correcting MCS improves post-merge fidelity and downstream performance.
>
> We firmly argue that addressing MCS is essential for reliable, data-efficient model merging when leveraging input data. By following a similar closed-form solution to RegMean, we directly address the root cause of failure for this widespread methodology and provide large accuracy gains: **+17\% accuracy on average across all benchmarks**.
>
> Secondary contributions, theoretically justified and empirically supported by ablation studies, include:
> - We incorporate our proposed task–layer importance measure int a modified regression objective, which allows us to increase performance even further.
> - We use only a small number of **balanced** examples per task (while Regmean uses all tasks' data), cutting execution time while preserving performance.
>
> **Baseline performance and implementation details.** $~~$ The discrepancy with prior results stems primarily from the experimental setting: while the original TA/TIES/DARE/Consensus/LiNeS results were reported under **full-rank fine-tuning**, all methods in our study operate on **LoRA-tuned modules**. This is in line with prior published works with the same experimental setup (Stoica et al., ICLR 2025; Panariello et al., NeurIPS 2025). Under LoRA, parameter-space task vectors are lower rank and the induced subspaces differ, which can degrade the absolute performance of methods designed for full-rank updates. This aligns with the findings in (Wang et al., ICLR 2025; Tang et al., ICML Workshop 2025) and explains why some variants perform similarly to TA in our tables.
>
> For all methods, we strictly followed the protocols from the original papers, and performed a **linear hyperparameter search** on the validation set around the recommended values. Our full code is provided in the supplementary material; we will add a per-baseline hyperparameter table and configuration details in the manuscript for transparency.
>
> **LoRA vs. full-rank results.** $~~$ We additionally include a **standard fine-tuning** evaluation (no LoRA) in the following. Results confirm the effectiveness of CoM and will be integrated into the manuscript as suggested.
>
> |**Method**| **Norm (B/32)**|**Abs (B/32)**|**Norm (L/14)**|**Abs (L/14)**|
> |-|:-:|:-:|:-:|:-:|
> TA | 76.5 | 70.8 | 88.7 | 84.9 |
> TIES | 81.2 | 75.1 | 90.8 | 86.9 |
> Consensus TA | 81.4 | 75.0 | 90.2 | 86.3 |
> FisherAVG | 73.8 | 68.3 | 85.9 | 82.2 |
> RegMean | 77.6 | 71.8 | 87.5 | 83.7 |
> Loc-and-Stitch | 86.3 | 79.9 | 90.4 | 86.5 |
> AdaMerging++ | 87.6 | 81.1 | 95.1 | 91.0 |
> ProDistill | 92.9 | 86.0 | 96.1 | 91.9 |
> TSV | 92.8 | 85.9 | 97.2 | 93.0 |
> **CoM (ours)**   | **94.8** | **87.7** | **97.8** | **93.6** |
>
> **Missing methods in computational cost comparison.** $~~$ We will add TA, TIES, DARE, Consensus, and LiNeS to the cost comparison in Section C. Their omission was for plot readability only.
>
> **Samples used for Gram matrices.** $~~$ We compute Gram matrices from the **validation** split. Other baselines use the same validation split for their linear hyperparameter search; we use it solely to estimate Gram statistics. We **never** use test samples for merging or tuning. Using the training split yields similar accuracy; we preferred using validation to fully comply with the experimental protocol.

---

> ### Author Response · Authors · 2025-11-24
>
> **Merging non-linear layers.** $~~$ If parameterized non-linear modules are present and fine-tuned, we merge them by **element-wise averaging across tasks**, as in RegMean, and then **recompute activations** with the merged model. In transformers, these include normalization layers and learned tokens/embeddings.
>
> **There are attention layers before the linear layers in the same residual block.** $~~$ Your point is correct: within a Transformer block, the attention sublayer precedes the MLP sublayer. However, attention projections (Q, K, and V matrices) are also linear layers, so they are merged with the same closed-form rule, and their activations are recomputed after each step to avoid merging covariate shift within the same residual block as well. We will clarify this in the manuscript.
>
> ### **References**:
>
> - Panariello et al., Accurate and Efficient Low-Rank Model Merging in Core Space. In NeurIPS, 2025.
>
> - Stoica et al., Model merging with SVD to tie the Knots. In ICLR, 2025.
>
> - Wang et al., 2025. LoRA-Pro: Are Low-Rank Adapters Properly Optimized? In ICLR, Spotlight.
>
> - Tang et al., 2025. LoRA Merging with SVD: Understanding Interference and Preserving Performance. In ICML Workshop.

---

> > ### Comment · Reviewer_1dpG · 2025-11-26
> >
> > Thanks to the authors for the additional clarifications and experiments in the rebuttal process. Given the validated performance gains on the full-rank finetuning setting, I have updated my final score to 6.

---

### Official Review · Reviewer_qsTg · 2025-11-01

**Soundness:** 2
**Presentation:** 3
**Contribution:** 3
**Rating:** 6
**Confidence:** 4

**Summary:**

The paper proposes an algorithm to perform model merging on transformer based models fine-tuned on different tasks. The main contribution proposed is to take into account the order of the merging to avoid error propagation throughout the layers, identified as covariance shift in feature space by the authors. The proposed solution is to merge one layer at a time from the initial layer to the last, instead of considering merging each layer independently and in parallel. The merging strategy is based on (Jin et al 2022) where preactivation features on each task (i.e. inner products between weights and features)  are approximated by a shared weight matrix on feature space, which has a closed form solution. The author propose further to weight the contribution of each task using gram matrix statistics of the features. Experiments are performed on vision and text benchmarks, showing good improvements on tested dataset.

**Strengths:**

- **Simple and effective contribution** Taking into account the propagation of error throughout layers in the merging, is a simple, well justified by the and very effective modification according to the result presented.

- **Strong measured performance**: performance on the models and datasets tested is consistently stronger with respect to the baselines tested.

- **Comprehensive experiments**: the paper compares with 10+ baselines on the vision and text domain in the merging experiments, showing strong performance and good coverage , with the only exception of few unmentioned works (see weakness section)

- **Ablations**: ablations experiments are useful to characterize the contribution of each component of the method proposed, showing the large impact of correcting the covariate shift between layers.

**Weaknesses:**

- **Task similarity coefficient**: according to what written in the paper the task similarity coefficient is computed on the normalized Gram   $XX^T$ matrices, by counting the magnitude of the off-diagonal elements. The authors support this choice claiming that larger correlations between features indicate higher distance to the pretrained features Gram matrix and hence, higher task complexity. It is not clear to me why this is the case: why near orthogonal samples in the fine-tuned would imply good task generalization? Distance is not directly measure w.r.t. Gram Matrices son the pretrained model as far as I understand. Moreover, isn't this meant to be computed on the covariance matrix $X^TX$, to test orthogonality of the features rather than samples? Please clarify.

- **Missing related works**: I think there are some missing related works [a,b] from previous methods that take into account layerwise dependency and would matter to discuss and compare to: especially [a] which proposes to merge task adaptively by weighting the contribution of each layer, being related to the layer ordering approach considered in the paper and improving over (Jing et al 2022).


- **Testing on out of distribution settings** I think the lack of OOD experiments is a missed opportunity to confirm the performance of the method. this could be done by simply leaving out some of the tasks and testing the accuracy them, similarly as done in (Jin et al 2022) or [a].

- **Missing absolute performance of the models**. I believe the paper would benefit from reporting the absolute performances of the fine-tuned model and base models on the different task in the Appendix as a reference to understand how well these perform.


_[a] Yang, Enneng, et al. "Adamerging: Adaptive model merging for multi-task learning.", ICLR 2024_

_[b] Xu, Jing, Jiazheng Li, and Jingzhao Zhang. "Scalable Model Merging with Progressive Layer-wise Distillation.”, ICML 2025_

**Questions:**

- **Task similarity coefficient ablation**: in the text of the paper authors specify that the similarity coefficient is computed on actuation of the merged model, for efficiency reason. Would the similarity coefficients computed on the finetuned models  activations be highly correlated to the one obtained on the merge models?


- **Robustness to sampling**: I think it would be useful to get variance across different sampling for each tasks, by e.g. averaging and reporting std on the results in Table (REF TABLE) across different samplings.

- **Balanced sampling across tasks**: when the tasks have different complexity, could a different number of samples be needed?

- **Scalability** What's the scalability of the method as opposed to averaging based methods? could it be employed e.g. on larger LLMs?

- **Merging LoRA weights ablation** Did you try to merge the LoRA parameters of the fine-tuned models instead of the original weights? (Assuming they have the same bottleneck dimensions across tasks)

- **Suggestion**: I would personally move the experiment on covariate shift earlier in the paper as a motivation to introduce the approach. It would be also useful to add the same measure on the merged model accounting for the order of layers to show that the covariance shift is lower in that case.

---

> ### Author Response · Authors · 2025-11-24
>
> ### **Task similarity coefficient**
> > Why near orthogonal samples in the fine-tuned would imply good task generalization?
>
> As you rightly pointed out, we do not compare pretrained and fine-tuned Gram matrices. Instead, we quantify inter-feature correlation **using a single model**: the pretrained one.
>
> We argue that, for in-distribution inputs, pretrained representations are approximately decorrelated, as they are learned from large and diverse datasets that capture broad, general-purpose structure rather than task-specific patterns. Therefore, high off-diagonal correlations in the input gram matrix $G$ indicate that the task concentrates on a narrow subspace of the original data distribution, and has diverged far from the pretraining point. Intuitively, when a task relies on only a few feature directions, those features become more correlated, revealing that the model is focusing on a narrower part of the pretrained representation space. This aligns with prior results showing that decorrelation improves generalization (Cogswell et al., 2015; Morcos et al., 2018) and with our empirical findings in Section 5 (feature correlation).
>
> The pretrained model would be the gold-standard reference, because its trained on a general-purpose large dastaset. However, computing correlations there would cost additional forward passes, so we instead use the merged model to keep CoM fast and scalable. The merged model behaves similarly because it is task-agnostic and provides a balanced representation across all tasks. To support this claim, we include an additional ablation study (Appendix E.2) that compares the pretrained and merged model explicitly.
>
> > Isn't this meant to be computed on the covariance matrix, to test orthogonality of the features rather than samples?
>
> We agree, orthogonal samples do not imply generalization; inter-feature correlation does. Let us clarify in the following *(normalization is omitted for ease of notation)*.
>
> Considering $X_i^l\in\mathbb{R}^{d\times n}$ the activations for layer-$l$ and task $i$, being $d$ the feature dimension ($X$ has shape *features $\times$ samples*). We form a **feature-feature** Gram matrix:
> $$
> G_i^l = X_i^l {X_i^l}^\top\in\mathbb{R}^{d\times d}.
> $$
> Here, $(G_i^l)_{pq}$ denotes the correlation between the generic features $p$ and $q$, **not** between samples. A larger off-diagonal $\ell_1$ norm indicates stronger inter-feature dependence.
>
> ### **Discussion on the suggested related works**
> >I think there are some missing related works [AdaMerging, ProDistill] that take into account layerwise dependency and would matter to discuss and compare to.
>
> We have expanded our related works discussion with the comments below, and included both methodologies in the experimental section.
>
> **AdaMerging** learns task-wise (or layer-wise) scalar coefficients via gradient descent, optimizing for all layers jointly, to minimize the entropy of the final predictions on unlabeled test data. Similarly, **ProDistill** learns layer-wise scalar coefficients but does that *independently* for each layer, and with the different objective of minimizing the $\ell_2$ norm between fine-tuned and merged models activations (similar to our proposed method, and RegMean).
>
> Instead, our **Chain of Merges** follows RegMean in its objectve, minimizing the *squared* $\ell_2$ norm between the activation of the merged model and the original fine-tuned models. Rather than updating a single scalar per layer via gradient descent, we solve for **all layer parameters** leveraging the optimal closed-form solution, and weight each task with an importance measure derived from feature correlation.
>
> **On layer-wise dependency.** $~~$ While ProDistill optimizes per-layer coefficients independently, AdaMerging optimizes them jointly across layers. This naturally mitigates distribution shifts that can arise during parameter merging. Our approach is related but different: we explicitly handle such shifts by updating activations and compute the merged parameters in closed form, rather than learning multiplicative coefficients through joint optimization.
>
> ### **Experimental comparison with the suggested related works**
>
> |**Method**|**ViT-B/32 Norm**|**ViT-B/32 Abs**|**ViT-L/14 Norm**|**ViT-L/14 Abs**|
> |-|:-:|:-:|:-:|:-:|
> AdaMerging++ | 87.6 | 81.1 | 95.1 | 91.0 |
> ProDistill | 92.9 | 86.0 | 96.1 | 91.9 |
> **CoM (ours)**   | **94.8** | **87.7** | **97.8** | **93.6** |
>
> In this setup $-$ now included in our revised manuscript $-$ we evaluate CoM against the proposed methods. Although the performance gains are smaller, CoM remains state-of-the-art when assessed **in the original evaluation protocol** used by these methods (traditional fine-tuning with checkpoints from Ilharco et al., 2023, "Editing models with task arithmetic").

---

> > ### Author Response · Authors · 2025-11-24
> >
> > ### **Out-of-distribution evaluation**
> > > I think the lack of OOD experiments is a missed opportunity to confirm the performance of the method.
> >
> > We thank the reviewer for the insightful suggestion.
> > Following AdaMerging (Table 3), we conduct two additional experiments, each merging on $6$ tasks (which we refer to as in-distribution, ID) and evaluating on the remaining $2$ (out-of-distribution, OOD). Our method attains state-of-the-art results on ID tasks in all cases, while AdaMerging is strongest on OOD. Notably, CoM is consistently the runner-up to AdaMerging variants on OOD. This suggests that, although CoM is less effective than AdaMerging at generalizing to unseen tasks, it remains highly competitive overall. These results are included in the new revision (appendix).
> >
> > **Seen $\rightarrow$ Unseen (ViT-B/32, 6 $\rightarrow$ 2 tasks)**
> > ID: SUN397, Cars, RESISC45, DTD, SVHN, GTSRB
> > OOD: MNIST, EuroSAT
> > |Method|Avg. accuracy, ID|Avg. accuracy, OOD|
> > |-|:-:|:-:|
> > |Task Arithmetic|70.6|61.7|
> > |Ties-Merging   |69.3|59.6|
> > |AdaMerging     |77.4|**70.0**|
> > |AdaMerging++   |78.0|68.7|
> > |Consensus      |72.8|59.5|
> > |Lines          |73.6|59.2|
> > |TSV            |76.8|59.3|
> > |CoM (ours)     |**81.1**|64.5|
> >
> > **Alternative split**
> > ID: SUN397, Cars, GTSRB, EuroSAT, DTD, MNIST
> > OOD: RESISC45, SVHN
> > |Method|Avg. accuracy, ID|Avg. accuracy, OOD|
> > |-|:-:|:-:|
> > |Task Arithmetic|73.9|51.1|
> > |Ties-Merging   |71.8|53.9|
> > |AdaMerging     |80.3|55.5|
> > |AdaMerging++   |80.8|**58.5**|
> > |Consensus      |73.9|53.3|
> > |Lines          |75.9|52.3|
> > |TSV            |79.0|51.9|
> > |CoM (ours)     |**83.4**|56.2|
> >
> > > I believe the paper would benefit from reporting the absolute performances of the fine-tuned model and base models
> >
> > Performance of the base model (zero-shot) and fine-tuned models (Individual FT) have been included in the revised manuscript.
> >
> > ### **Questions**
> > **Task-similarity coefficients ablation.** $~~$ To keep coefficients comparable across tasks and layers, we compute them on a single, task-agnostic reference model. Using task-specific fine-tuned models would not quantify each task’s out-of-distribution shift; it would primarily reflect how well each model fits its own task (greater feature decorrelation would indicate better in-task generalization, not cross-task similarity). Moreover, computing coefficients on different fine-tuned references would also put values on different scales, breaking comparability. The ideal reference is instead the zero-shot (pretrained) model. In practice, we use the merged model, which is approximately task-agnostic and whose input Gram matrices are already available in our pipeline. To justify this claim, we compare the similarity measure computed with the merged model to that from the zero-shot (pretrained) model. In the revised manuscript (Appendix), an additional ablation shows that the two measures are highly correlated, supporting our approximation.
> >
> > **Robustness to sampling.** $~~$ We omit error bars because they are negligible: the mean standard deviation across random samplings is $0.017\%$, with a maximum of $0.037\%$ (on ViT-L/14). These consistently low values indicate that CoM’s performance is largely insensitive to the particular data sampling.
> >
> > > What's the scalability of the method as opposed to averaging based methods? could it be employed e.g. on larger LLMs?
> >
> > **Scalability.** $~~$ Our method is more demanding than averaging-based approaches, due to a per-layer matrix inversion. The computational cost scales as $O(\text{dim}^3)$ **per layer** (i.e., $O(L\,\text{dim}^3)$ overall for $L$ layers), while memory scales linearly with data as $O(N\,\text{dim})$. In practice, these operations map well to optimized linear algebra kernels; we have already applied the method to **LlaMA-3 8B**, and the same recipe extends to larger LLMs with batching and (if needed) distributed compute. Figure 5 (appendix) shows the computational cost of our model on Llama-3 8B.
> >
> > **Balanced sampling across tasks.** $~~$ We have not explored per-task sample budgets, as in our experiments a small fixed number of samples already worked well (see Table 4, main paper). We thank the reviewer for the suggestion; we agree that leveraging a smaller sample count for simpler tasks is a promising idea and could lower the number of needed samples even more. We will consider it in future experiments and discussion.

---

> > > ### Author Response · Authors · 2025-11-24
> > >
> > > ### **Merging LoRA**
> > > >Did you try to merge the LoRA parameters of the fine-tuned models instead of the original weights?
> > >
> > > Merging LoRA residuals or full weights is equivalent in our framework, but going back to low rank merged matrices is generally not possible.
> > >
> > > Let $W_{\mathrm{FT},i}=W_{\mathrm{PT}}+B_iA_i$ with $\mathrm{rank}(B_iA_i)\le R$ be the fine-tuned layer at hand, for all models $i=1,2,\dots,N$. Because the LoRA residuals $\[B_i A_i\]\_{i=1}^N$ add linearly to the pretrained weights $W_{\mathrm{PT}}$, the shared $W_{\mathrm{PT}}$ cancels out inside the $\ell_2$ squared norm in Equation 1\. Formally:
> > >
> > > Let $W_{\mathrm{FT},i}=W_{\mathrm{PT}}+B_i A_i$ with $\mathrm{rank}(B_i A_i)\le R$, for $i=1,\dots,N$.
> > >
> > > Consider the least-squares merge (Eq. 1):
> > >
> > > $\hat W = \arg\min_{W}\sum_{i=1}^N \big\|W\_{\mathrm{FT},i}-W\big\|_2^2 = \arg\min\_{W}\sum\_{i=1}^N\big\|(W\_{\mathrm{PT}}+B_i A_i)-W\big\|\_2^2.$
> > >
> > > By translation invariance of the $\ell_2$ norm,
> > >
> > > $\sum\_{i=1}^N\big\|(W\_{\mathrm{PT}}+B_iA_i)-W\big\|_2^2=\sum\_{i=1}^N\big\|B_i A_i-(W-W\_{\mathrm{PT}})\big\|_2^2.$
> > >
> > > Let $Z := W - W_{\mathrm{PT}}$. Then
> > >
> > > $\hat Z = \arg\min_{Z}\sum\_{i=1}^N \big\|B_iA_i - Z\big\|_2^2,\qquad\hat W = W\_{\mathrm{PT}} + \hat Z .$
> > >
> > > Hence, merging $\{W_{\mathrm{PT}}+\[B_iA_i\]}\_{i=1}^N$ is equivalent to merging the residuals $[B_iA_i]\_{i=1}^N$ and adding back the common $W_{\mathrm{PT}}$.
> > >
> > > In both cases, $\mathrm{rank}(\Delta W_M)\;\le\;\sum_{i=1}^N \mathrm{rank}(B_iA_i)\;\le\; N R$, so the merged residual $\Delta W_M$ cannot, in general, be factored back into a single LoRA pair $(\tilde B,\tilde A)$ with rank $R$ without discarding information, unless task subspaces align.
> > >
> > > **Suggestion: also report MCS for CoM.** In our proposed CoM, merging covariate shift is zero by construction: after each layer is merged, we recompute activations with the merged weights, so subsequent merges use matched inputs and accumulate no shift whatsoever.

---

> > > > ### Comment · Reviewer_qsTg · 2025-11-25
> > > > **Response to rebuttal**
> > > >
> > > > I thank the authors for their response and for the additional experiments performed. Their rebuttal addresses most of my concerns, therefore I recommend the paper for acceptance.

---

### Author Response · Authors · 2025-11-24

We followed the insightful reviewers' feedback and revised the paper to clarify scope and strengthen our empirical evaluation, showing that **CoM confirms its state-of-the-art performance on full fine-tuning**. Specifically, we incorporated:

- **[SXdo, 1dpG, Pa3c]** Full fine-tuning experiments on ViT-B/32 and ViT-L/14; CoM remains SOTA under this protocol.
- **[SXdo, Pa3c, qsTg]**Absolute accuracies for all experiments, with base (zeroshot) and individual models performance.
- **[qsTg, Pa3c]** Comparison with AdaMerging, ProDistill, and Localize and Stitch in text and experiments, clarifying why EMR-merging (input-dependent computation at inference) is out-of-scope for our scenario.
- **[qsTg]** OOD evaluation following AdaMerging, ablation study comparing merged-model vs pretrained references, and LoRA merging equivalence derivation, showing equivalence between merging full weights $(W_{\mathrm{PT}}+B_iA_i)$ and residuals $(B_iA_i)$.

We thank all reviewers for recognizing strengths such as clarity, the well-motivated merging covariate shift, and strong empirical results.

---

### Author Response · Authors · 2025-12-02
**Summary for the Area Chair: Rebuttal & Changes**

We sincerely thank all reviewers for their insightful and constructive comments, and we equally appreciate the Area Chair for dedicating their time and attention to our paper, which was already **acknowledged by all reviewers as clearly written and empirically solid**, and by several reviewers as **presenting an interesting idea supported by a well-motivated contribution**.

### What happened in the rebuttal/discussion
- Reviewer **qsTg** and **1dpG** appreciated our rebuttal, noting that our response addressed their primary concerns. **Both reviewers increased their scores in favor of acceptance**, rating the paper an 8 and a 6, respectively.

- Reviewer **Pa3c** found our conceptual and empirical explanations satisfactory, but requested a **computational and memory footprint comparison** to dynamic merging methods; we provided it in our last response but the discussion was stopped.

- Reviewer **SXdo** initially leaned toward rejection but did not take part in the discussion phase. Nonetheless, we fully addressed their two main concerns by **adding full fine-tuning results** and **reporting absolute accuracies** throughout the paper.

Overall, three out of four reviewers were satisfied with our clarification and additional analyses. Among these, reviewer Pa3c requested one specific additional comparison, which we provided, while the fourth reviewer (SXdo) did not participate before the discussion was stopped.

### **Revision changes:**
- [qsTg, 1dpG, Pa3c, SXdo] **Added full-fine-tuning (FFT) results** in the main paper (Table 2). Added a comparison with *AdaMerging*, *ProDistill*, and *Localize and Stitch*. Our proposed method (CoM) remains SOTA in all settings.
- [qsTg, Pa3c, SXdo] **Reported absolute accuracies everywhere**, along with base (zeroshot) and individual models performance. CoM also provides SOTA results under this metric.
- [1dpG, Pa3c] **Clarified the compute & memory analysis of the merging step.** Figure 5 shows that CoM’s merging-time FLOPs and peak memory match strong regression/SVD baselines; we also clarified the evaluation protocol (static single-model inference, no routing or task IDs) and added a discussion of dynamic merging techniques with a full computational and memory footprint analysis in the rebuttal.
- [qsTg] **Additional experiments.** We (i) added OOD evaluation following the AdaMerging protocol; (ii) motivated the use of the merged model to compute our task-importance metric; and (iii) derived the LoRA-merging equivalence between merging full weights and residuals.

---

### Meta-Review · Area_Chair_zpGv · 2026-01-05

**Summary:**

The paper proposes Chain of Merges (CoM), a layer-wise model merging method that sequentially merges layers while recomputing activations to mitigate what the authors call “merging covariate shift” (MCS). Reviewers agree that the empirical results are generally strong within the reported setup, and that the idea of explicitly accounting for inter-layer dependencies is interesting. However, several key concerns remain. First, the novelty is viewed as modest: CoM is perceived largely as an incremental refinement of RegMean and related regression-based merging approaches, with the main change being to run the regression on merged-model activations rather than original fine-tuned activations. Second, there are scope and fairness issues in the experimental validation. Much of the initial evaluation was limited to LoRA-fine-tuned models, while claims and comparisons were framed as general model merging; this raised questions about the validity of “state-of-the-art” claims against baselines developed and tuned in full fine-tuning settings. Third, reviewers expressed concerns about baseline performance and fairness of comparisons, notably the gap between TSV’s reported performance in its original paper and the much lower TSV performance here, as well as the absence or limited discussion of strong recent methods (e.g., EMR-Merging, Twin-Merging, WeMoE) that use examples and/or conditional inference. Finally, some reviewers raised missing or underspecified experimental details (number and source of examples, treatment of non-linear layers, compute/memory costs vs baselines), and questioned whether the evaluation scale (number of tasks, OOD settings) and protocol were sufficient to fully support the paper’s broader claims.

**Reviewer Concerns:**

The rebuttal and subsequent revisions substantially improved the clarity and scope of the paper, and several concerns were meaningfully addressed:

- The authors added full fine-tuning (FFT) experiments on ViT-B/32 and ViT-L/14, showing that CoM remains strong or state-of-the-art under a standard full-parameter fine-tuning protocol.

- They reported absolute accuracies alongside normalized scores for all experiments, clarifying that the method’s gains are not an artifact of normalization.

- They specified the number and source of examples used for Gram matrix estimation for each backbone and clarified that examples come from the validation set; they also provided ablations on sample budgets and correlations between using the merged vs pretrained model as reference.

- They expanded the related work and added experimental comparisons to AdaMerging, ProDistill, and other static merging baselines, as well as OOD evaluations following the AdaMerging protocol.

- They clarified the treatment of non-linear and attention layers, explaining that all linear layers (including attention projections) are merged with the closed-form rule, while other parameterized non-linear modules are averaged and activations recomputed.

- They provided detailed compute and memory analyses for CoM against a range of static and dynamic methods, and clarified the intended “single static merged model, no routing, no task IDs” evaluation protocol.


However, some core concerns remain outstanding:

- Fairness and completeness of comparisons. Reviewer Pa3c, in particular, remains unconvinced about the fairness of comparisons to TSV and the decision not to include direct numerical comparisons to dynamic or conditional methods such as EMR-Merging, Twin-Merging, and WeMoE, especially given that CoM also uses examples and validation data. The authors provide a detailed argument about different inference regimes (static vs conditional), but this does not fully resolve the concern that, in practice, CoM competes for similar use cases and should be positioned more carefully against these methods.

- Scope and evaluation breadth. While FFT experiments and OOD evaluations were added, the overall experimental scope (task counts, settings) is still narrower than in some recent works, and the paper’s general claims about “state-of-the-art” model merging remain somewhat stronger than what the evidence can unambiguously support across all regimes.

- Positioning within the broader merging landscape. Even with the improved discussion, there remains some ambiguity about where CoM sits relative to the growing body of methods that use examples, conditional routing, or modular experts. This affects how compelling the overall contribution appears, particularly to the more critical reviewers.


Given these outstanding issues, I do not think the rebuttal fully resolves the main concerns for a clear acceptance.

**Reviewer Scores:**

Reviewer qsTg initially rated the paper as marginally above threshold and, after the rebuttal and additional experiments, explicitly stated that most concerns were addressed and raised their score to an 8 and recommended acceptance. In a fully active discussion, I would expect qsTg to maintain this higher, positive score.

Reviewer 1dpG started slightly negative (4) due to concerns about modest novelty, unexpectedly low baseline performance, and missing implementation/compute details. The rebuttal provided FFT results, clarified LoRA vs full-rank behavior, and expanded cost and implementation details. Reviewer 1dpG explicitly acknowledges these improvements and updated their score to 6. With full participation in the discussion, I expect this reviewer to keep a weak-accept / borderline-positive score, not move to a clear reject.

Reviewer Pa3c was initially marginally below threshold (4), raising concerns about missing example details, the lack of absolute accuracies, limited scale, and missing comparisons (especially EMR-Merging and other example-using methods). The rebuttal addressed example counts, absolute accuracies, FFT results, and added more discussion and analysis of dynamic merging methods, but Pa3c explicitly states that concerns about fairness of comparisons, particularly vs TSV and example-using dynamic methods, remain unresolved, and did not change their score. Even after full discussion, I expect Pa3c to remain at a marginally-below-threshold (4) assessment.

Reviewer SXdo gave a clear reject (2), focusing on the misleadingly general framing given the initial LoRA-only experiments, invalid SOTA claims under that framing, and the absence of absolute accuracies. The rebuttal directly addresses these points by adding FFT experiments and absolute accuracies, and clarifying the scope and protocol. SXdo did not re-engage in the discussion, so we do not have an updated score. Based on the nature of their concerns and the rebuttal, I could imagine SXdo softening somewhat (e.g., to a 4) if fully engaged, but there is no strong evidence that they would move into the acceptance range; the fundamental skepticism about the strength and breadth of the contribution would likely remain.

In summary, after rebuttal and discussion, the likely final landscape is: one strong accept (8), one weak accept (6), and two marginal/negative scores (4 and 2, or possibly 4 and 4). Given the remaining doubts about novelty, fairness of comparisons, and the breadth and positioning of the contribution, I side with the more critical assessments and recommend rejection.

---

### Decision · Program_Chairs · 2026-01-26

Reject